# Laser interstitial thermal therapy and adjuvant pembrolizumab in recurrent high-grade astrocytoma: a Phase 1/randomized Phase 2b trial

Immune checkpoint inhibitors (ICIs) show minimal efficacy in recurrent high-grade astrocytoma (rHGA). Laser interstitial thermal therapy (LITT), a minimally invasive cytoreductive approach, may prime rHGA for ICI response. A phase 1/randomized phase 2b trial (ClinicalTrials.gov: NCT02311582) was designed to test pembrolizumab in combination with LITT in patients with rHGA. Nine patients were enrolled in the phase I dose-escalation lead-in study. No dose-limiting toxicities were observed and 200 mg of pembrolizumab every three weeks was determined as the recommended phase 2 dose. The phase 2b study was initially designed to randomize (up to 45) patients 1:1 to either LITT followed by pembrolizumab (LITT + PEM) or non-LITT surgery followed by pembrolizumab (NLS + PEM). Phase 2's primary endpoint was progression-free survival (PFS); secondary endpoints included overall survival (OS), safety, and immune signature. After 21 patients, based on an independent Data and Safety Monitoring Committee request of unscheduled interim review of accumulating efficacy data, randomization stopped as benefit from NLS + PEM appeared limited, and the subsequent 24 patients received LITT + PEM. The pre-specified study endpoints were achieved. Among 39 per-protocol patients, LITT + PEM (n = 33) improved median OS (11.8 versus 5.2 months) and 18-month survival (42% versus 0%) compared to NLS + PEM (n = 6) (hazard ratio [HR] 0.17; 95% confidence interval [CI], 0.06–0.49; P = 0.0002). Median PFS was longer in LITT + PEM (4.5 versus 1.6 months; HR 0.21; 95% CI, 0.08–0.56; P = 0.0006). In an intent-to-treat sensitivity analysis (n = 21), OS (HR 0.29; 95% CI, 0.10–0.88) and PFS (HR 0.30; 95% CI, 0.10–0.87) again favored LITT + PEM (n = 13). Treatment was well tolerated. LITT activated non-classical monocytes, and pembrolizumab unleashed CD8+ T cell proliferation, clonal expansion, and coordinated memory T-cell responses. Overall, LITT + PEM is safe and may overcome rHGA immunosuppression to generate anti-tumor immunity.

✉e-mail: mchheda@wustl.edu; alberthkim@wustl.edu; leuthardte@wustl.edu; david.tran@med.usc.edu

Recurrent HGA, including glioblastoma (GBM) and *IDH*-mutant World Health Organization (WHO) grade 3 and 4 astrocytoma, remains one of the greatest challenges in neuro-oncology. After maximal resection, radiotherapy, and temozolomide chemotherapy, salvage options for rHGA offer only modest benefit, and prognosis is dismal[1]. These realities highlight the urgent need for new therapies that can meaningfully extend survival and enhance quality of life for these patients.

ICIs, such as the anti-PD-1 antibodies pembrolizumab and nivolumab, have transformed care in many solid tumors by unleashing cytotoxic T-cell responses[2]. Yet, in HGA, ICIs, whether added to frontline chemoradiation or used alone or combined with the salvage therapy bevacizumab in recurrence, have failed to extend survival beyond these established treatments[3–7]. Resistance is thought to stem from the blood–brain barrier (BBB), low tumor neoantigen burden, and a highly immunosuppressive tumor microenvironment (TME) that limits CD8+ T-cell infiltration and function[8]. More recently, neoadjuvant ICIs given before maximal resection were shown to transiently "heat up" the TME by boosting tumor-infiltrating lymphocytes, TCR diversity, and peripheral T-cell clonal expansion, but this has not consistently translated into a survival advantage over adjuvant ICI alone in rHGA[9–12]. These mixed results underscore the ongoing challenge of stimulating durable anti-tumor immunity in rHGA, where both intrinsic tumor features and limited neoantigen availability that is further depleted following surgical resection remain major obstacles. Despite these disappointing clinical results, PD-L1 is detectable in a substantial subset of high-grade astrocytoma and tends to increase with higher WHO grade and PTEN loss, a common alteration in these tumors that is linked to immune evasion[13–17], providing a biological rationale for continued interest in evaluating PD-1/PD-L1 blockade in this setting.

LITT is a minimally invasive, MRI-guided thermal ablation technique for brain tumors[18–22]. It is used to achieve effective cytoreduction with lower surgical risk and has yielded survival outcomes comparable to open resection[23]. Moreover, LITT has been shown to induce transient BBB disruption, particularly in the peritumoral region, thereby enhancing CNS penetration of systemic agents that would otherwise be excluded[21,24,25]. Accordingly, the National Comprehensive Cancer Network guidelines have added LITT as a cytoreductive option for poor surgical candidates[26]. Beyond direct tumor debulking, the ablative injury appears to remodel the peri-tumoral TME by continuously liberating intracranial antigens, driving peripheral tumor-specific immune activation that peaks weeks to months later[21,27,28]. Leveraging these complementary mechanisms—antigen release, BBB disruption, and local immune priming—combining LITT with immune checkpoint blockade may overcome the profoundly immunosuppressive milieu and sparse neoantigen landscape of rHGA.

Here, we show in a Phase 1 lead-in and randomized Phase 2b study that LITT followed by pembrolizumab is safe and associated with improved PFS and OS compared with non-LITT surgery followed by pembrolizumab in rHGA. Longitudinal paired 5′ single cell RNA-seq (scRNA-seq) and TCR sequencing of peripheral blood mononuclear cells (PBMCs) reveals systemic immune remodeling after LITT and during PD-1 blockade, including early activation of non-classical monocytes and increased CD8+ T-cell proliferation with clonal expansion. Coordinated memory-to-effector CD8+ T cell responses characterize long-term survivors, whereas stagnant and exhausted phenotypes predominate in short-term survivors.

## Results

### Study design and patient demographics and baseline characteristics

Phase 1: Between May 2016 and May 2017, nine patients (four males) with bevacizumab- and immunotherapy-naïve recurrent WHO grade 3 (two patients) or grade 4 (seven patients) astrocytoma, unequivocal progression per the Response Assessment in Neuro-Oncology [RANO] criteria[29], and Karnofsky performance status (KPS) ≥ 60% were enrolled in a standard 3 × 3 Phase 1 study with LITT followed within 1 week by pembrolizumab at three dose levels of 100, 150, and 200 mg intravenously every 3 weeks (Fig. 1a). Adverse events were recorded using the Common Terminology Criteria for Adverse Events (CTCAE v. 4. 0). No dose limiting toxicities were observed at any of the three dose levels, and thus 200 mg every 3 weeks was the recommended Phase 2 dose (RP2D).

Phase 2: From January 2017 to December 2022, 45 patients with bevacizumab- and immunotherapy-naïve recurrent WHO grade 4 astrocytoma, unequivocal progression per RANO criteria[29], and KPS ≥ 60% were enrolled. The study originally specified 1:1 randomization between LITT + PEM and NLS + PEM, which proceeded for the first 21 patients (intent-to-treat [ITT] population: 8 → NLS + PEM; 13 → LITT + PEM). In mid-2019, emerging data demonstrating limited benefit from NLS + PEM[11] prompted an independent Data and Safety Monitoring Committee (DSMC)-requested, unscheduled interim review of accumulating efficacy data, which suggested a survival signal favoring LITT + PEM. Protocol Amendment 12 subsequently halted randomization and assigned all remaining enrollments (*n* = 24) to the LITT + PEM arm. The trial then continued as a single-arm LITT + PEM extension designed to test whether PFS exceeded a conservative historical benchmark set at 2.9 months, compared with the 2.4-month mPFS reported for NLS + PEM in the neoadjuvant pembrolizumab trial in rHGA[11]. Although upfront surgical debulking before LITT was permitted, in practice, no patient in either the randomized or single-arm expansion cohorts underwent resection prior to laser ablation. Patients with suspected secondary WHO grade 4 astrocytoma (prior grade 2–3 disease treated with chemoradiation and new imaging suggesting transformation to grade 4) were provisionally enrolled pending histopathological confirmation of grade 4 on biopsy or resection; unconfirmed cases were replaced.

In the ITT cohort, two patients randomized to NLS + PEM withdrew consent after randomization to receive off-trial LITT + PEM, and three patients randomized to LITT + PEM without confirmed grade 4 histology were treated off-trial with chemotherapy. Among the 24 patients enrolled after Amendment 12, one was excluded due to insurance denial for LITT and three for unconfirmed grade 4 disease (two of whom were subsequently lost to follow-up). Ultimately, 30 Phase 2 patients received LITT + PEM (10 randomized, 20 post-amendment). For the per-protocol analysis, we added three Phase 1 patients treated at the RP2D (dose level 3) who met Phase 2 eligibility, yielding 33 evaluable LITT + PEM and six evaluable NLS + PEM patients (*n* = 39 for per-protocol cohort; see Fig. 1a). The primary endpoint was PFS; secondary endpoints were OS, objective response rate, safety, and immunophenotyping, with MRI assessments per RANO at baseline and every 9 weeks (Fig. 1b).

In both the ITT and per-protocol cohorts, the LITT + PEM and NLS + PEM arms were well matched for key baseline characteristics, including age, sex, KPS, initial diagnosis, *MGMT* promoter methylation, *IDH* mutation status, prior recurrences, tumor location, and treatment center (Table 1). All patients had at least one prior recurrence, i.e., they were at second recurrence or beyond at trial entry, and several had reached a third (or greater) recurrence, underscoring the cohort's advanced disease stage. After the early closure of the NLS + PEM arm, we established a contemporaneous historical control (CHC) of 13 recurrent, wild-type-*IDH* (wt-*IDH*) GBM patients treated at Washington University with LITT followed by non-ICI therapies (temozolomide, lomustine, carboplatin, or bevacizumab; see Supplementary Fig. S1a and "Methods" for cohort creation). This CHC was intended to bolster our comparison and isolate the effects of LITT without ICI. The CHC cohort (all wt-*IDH*; 61.5% female; 53.8% *MGMT*-methylated) closely matched our 29 wt-*IDH* GBM patients in the LITT + PEM arm, although CHC tumors were disproportionately frontal and included one cerebellar and one corpus-callosal lesion (Table 1 and Supplementary Table S1).

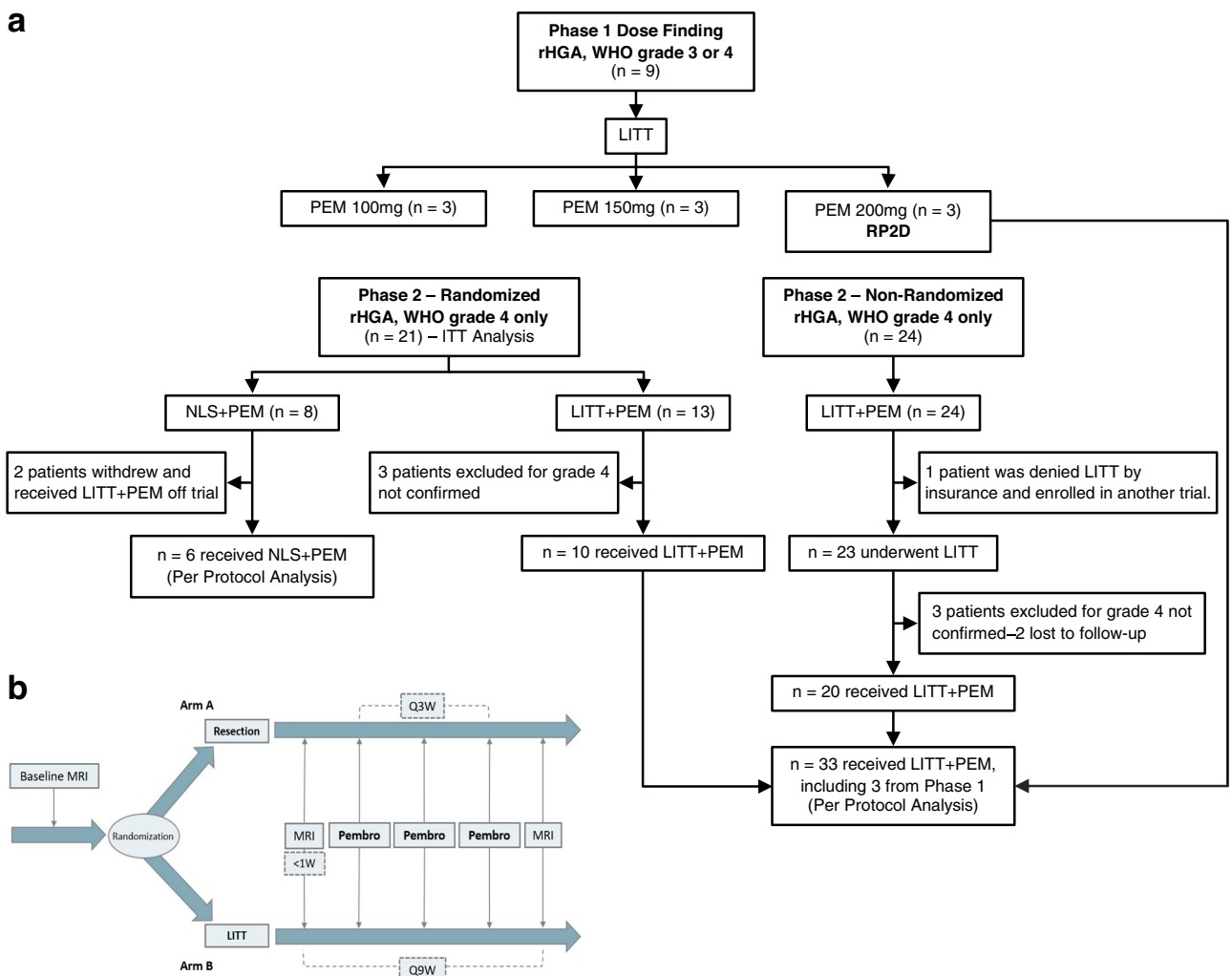

**Fig. 1 | Study design and enrollment plan. a** A CONSORT table detailing enrollments for Phase 1 and the randomized and non-randomized portions of Phase 2. Intent-to-treat (ITT) analysis included 21 randomized Phase 2 patients. Three Phase 1 patients who received LITT followed by the recommended Phase 2 dose (RP2D) of pembrolizumab and met Phase 2 criteria were combined with 30 Phase 2 LITT + PEM patients for the per-protocol analysis, compared to six NLS + PEM patients. **b** Treatment and monitoring schema for Phase 2. Pembrolizumab was administered within one week of LITT or NLS.

## Safety

The majority of study treatment-attributed adverse events were mild (grades 1–2), with the most common being elevated alanine aminotransferase (six patients; 13%), diarrhea (six patients; 13%), muscle weakness (five patients; 11%), pruritus (four patients; 9%), nausea (four patients; 9%), and arthralgia (four patients; 9%) (Table 2). Grade 3–4 events requiring hospitalization included diarrhea (two patients; 4%) and muscle weakness (two patients; 4%) and resolved with treatment. No grade 3–4 event occurred in more than two patients. There was no grade 5 events.

## LITT followed by pembrolizumab is associated with extended survival in rHGA

Summaries of OS and PFS for all Phase 1 and Phase 2 enrollments, along with best response rates and representative brain images for the 39-patient per-protocol cohort, are shown in Fig. 2. In the randomized ITT population, LITT + PEM achieved a median OS of 21.3 months with 54% OS18, compared to 6.7 months with 12.5% OS18 in the NLS + PEM arm (HR 0.29; 95% CI, 0.1–0.88; log-rank $P = 0.021$), and median PFS improved to 7.8 months versus 2.2 months (HR 0.30; 95% CI, 0.10–0.87; log-rank $P = 0.020$) (Fig. 3a). Because randomization was halted early and the trial subsequently continued with single-arm enrollment, these between-arm comparisons are exploratory. In the

per-protocol cohort, LITT + PEM extended median OS to 11.8 months with 42% OS18, versus 5.2 months with 0% OS18 (HR 0.17; 95% CI, 0.06–0.49; log-rank $P = 0.0002$), while median PFS increased to 4.5 months compared with 1.6 months (HR 0.21; 95% CI, 0.08–0.56; log-rank $P = 0.0006$) (Fig. 3b). Since the ITT population includes five randomized patients who did not complete protocol-defined treatment or lacked confirmed grade 4 histology and were excluded from the per-protocol analysis, differences in cohort composition and censoring likely contributed to the ITT median estimates, although no single extreme outlier drove the ITT survival curves. Finally, as a descriptive comparison to the conservative historical benchmark of 2.9 months specified for the amended single-arm extension, the per-protocol LITT + PEM cohort had an mPFS of 4.5 months (95% CI, 3.4–10.3), with the lower confidence bound exceeding 2.9 months.

Among wt-*IDH* GBM patients, LITT + PEM again outperformed NLS + PEM, with median OS of 11.1 versus 4.8 months (HR 0.16; 95% CI, 0.05–0.50; log-rank $P = 0.0003$) and PFS of 3.9 versus 1.7 months (HR 0.29; 95% CI, 0.10–0.81; log-rank $P = 0.01$) (Fig. 3c). Compared to the CHC treated with LITT and non-ICI agents, which was measured from the date of LITT rather than first pembrolizumab dosing as above, LITT + PEM maintained a significant survival benefit, with median OS of 11.3 versus 6.5 months (HR 0.27; 95% CI, 0.12–0.60; log-rank $P = 0.0006$) and PFS of 4.1 versus 2.1 months (HR 0.30; 95% CI, 0.14–0.65; log-rank

**Table 1 | Baseline demographic and clinical characteristics**

| Characteristics | ITT randomized treatment arms | | | | Per protocol treatment arms | | | | wt-IDH GBM treatment arms | | | |
|---|---|---|---|---|---|---|---|---|---|---|---|---|
| | LITT+PEM n=13 | NLS+PEM n=8 | Overall n=21 | P-value | LITT+PEM n=33 | NLS+PEM n=6 | Overall n=39 | P-value | LITT+PEM n=29 | NLS+PEM n=5 | Overall n=34 | P-value |
| Age, Median (IQR) | 60 (52,66) | 57 (49,64) | 59 (51,64) | 0.586 | 61 (49,67) | 53 (47,64) | 60 (49,67) | 0.436 | 61 (49,67) | 51 (47,66) | 61 (49,66) | 0.451 |
| Sex, n (%) | | | | 1.000 | | | | 1.000 | | | | 1.00 |
| Female | 7 (53.8) | 5 (62.5) | 12 (57.1) | | 16 (48.5) | 3 (50.0) | 19 (48.7) | | 14 (48.3) | 3 (60.0) | 17 (50.0) | |
| Male | 6 (46.2) | 3 (37.5) | 9 (42.9) | | 17 (51.5) | 3 (50.0) | 20 (51.3) | | 15 (51.7) | 2 (40.0) | 17 (50.0) | |
| KPS, Median (range) | 80 (70–100) | 80 (70–80) | 80 (70–100) | 0.30 | 80 (70–100) | 80 (80) | 80 (70–100) | 0.34 | 80 (70–100) | 80 (80–90) | 80 (70–100) | 0.49 |
| Diagnosis, n (%) | | | | 0.336 | | | | 1.000 | | | | 1.000 |
| Grade 3 rHGA | 3 (23.1) | 0 (0.0) | 3 (14.3) | | 0 (0.0) | 0 (0.0) | 0 (0.0) | | 0 (0.0) | 0 (0.0) | 0 (0.0) | |
| Grade 4 rHGA | 1 (7.7) | 1 (12.5) | 2 (9.5) | | 4 (12.1) | 1 (16.7) | 5 (12.8) | | 0 (0.0) | 0 (0.0) | 0 (0.0) | |
| GBM | 9 (69.2) | 7 (87.5) | 16 (76.2) | | 29 (87.9) | 5 (83.3) | 34 (87.2) | | 29 (100.0) | 5 (100.0) | 34 (100.0) | |
| IDH mutation, n (%) | | | | 1.000 | | | | 1.000 | | | | 1.000 |
| Wild-type | 11 (84.6) | 7 (87.5) | 18 (85.7) | | 29 (87.9) | 5 (83.3) | 34 (87.2) | | 29 (100.0) | 5 (100.0) | 34 (100.0) | |
| Mutant | 2 (15.4) | 1 (12.5) | 3 (14.3) | | 4 (12.1) | 1 (16.7%) | 5 (12.8) | | 0 (0.0) | 0 (0.0) | 0 (0.0) | |
| MGMT status, n (%) | | | | 0.211 | | | | 1.000 | | | | 0.734 |
| Unknown | 5 (38.5) | 1 (12.5) | 6 (28.6) | | 6 (18.2) | 1 (16.7) | 7 (17.9) | | 4 (13.8) | 1 (20.0) | 5 (14.7) | |
| Unmethylated | 1 (7.7) | 3 (37.5) | 4 (19.0) | | 11 (33.3) | 2 (33.3) | 13 (33.3) | | 11 (37.9) | 1 (20.0) | 12 (35.3) | |
| Methylated | 7 (53.8) | 4 (50.0) | 11 (52.4) | | 16 (48.5) | 3 (50.0) | 19 (48.7) | | 14 (48.3) | 3 (60.0) | 17 (50.0) | |
| Recurrence, n (%) | | | | 1.000 | | | | 0.600 | | | | 1.000 |
| 2nd Recurrence | 11 (84.6) | 7 (87.5) | 18 (85.7) | | 30 (91.0) | 6 (100.0) | 36 (92.3) | | 26 (89.7) | 5 (100.0) | 31 (91.2) | |
| ≥ 3rd Recurrence | 2 (15.4) | 1 (12.5) | 3 (14.3) | | 3 (9.0) | 0 (0.0) | 3 (7.3) | | 3 (10.3) | 0 (0.0) | 3 (8.8) | |
| Tumor location, n (%) | | | | 0.660 | | | | 0.590 | | | | 0.550 |
| Frontal lobe | 4 (30.8) | 4 (50.0) | 8 (38.1) | | 14 (42.4) | 3 (50.0) | 17 (43.6) | | 11 (37.9) | 2 (40.0) | 13 (38.2) | |
| Temporal lobe | 6 (46.1) | 3 (37.5) | 9 (42.9) | | 11 (33.3) | 3 (50.0) | 14 (35.9) | | 10 (34.5) | 3 (60.0) | 13 (38.2) | |
| Parietal lobe | 3 (23.1) | 1 (12.5) | 4 (19.0) | | 6 (18.2) | 0 (0.0) | 6 (15.4) | | 6 (20.7) | 0 (0.0) | 6 (17.7) | |
| Occipital lobe | 0 (0.0) | 0 (0.0) | 0 (0.0) | | 2 (6.1) | 0 (0.0) | 2 (5.1) | | 2 (6.9) | 0 (0.0) | 2 (5.9) | |
| Institution, n (%) | | | | 1.000 | | | | 0.091 | | | | 0.164 |
| FL | 7 (53.8) | 5 (62.5) | 12 (57.1) | | 14 (42.4) | 5 (83.3) | 19 (48.7) | | 12 (41.4) | 4 (80.0) | 16 (47.1) | |
| WU | 6 (46.2) | 3 (37.5) | 9 (42.9) | | 19 (57.6) | 1 (16.7) | 20 (51.3) | | 17 (58.6) | 1 (20.0) | 18 (52.9) | |

Data are median (IQR) or n (%).
P values were calculated using the Wilcoxon rank-sum test for continuous variables and Fisher's exact test for categorical variables.

**Table 2 | All study treatment-attributable adverse events by grade (> 5% frequency)**

| AE class disorders | Event | NLS + PEM (n = 6) | | LITT + PEM (n = 33) | |
|---|---|---|---|---|---|
| | | Grade <3 | Grade ≥3 | Grade <3 | Grade ≥3 |
| **General** | Fever | | | 2 | |
| | Fatigue | 1 | | 2 | |
| | Pain | | | 2 | |
| **Blood and lymphatic system** | Elevated aspartate aminotransferase | | | 1 | |
| | Elevated alanine aminotransferase | | 1 | 6 | |
| | Lymphopenia | | 1 | | |
| | Elevated alkaline phosphatase | | | 3 | |
| **Endocrine** | Hypothyroidism | | | 2 | |
| **Gastrointestinal** | Nausea | | | 4 | |
| | Diarrhea | | | 6 | 2[a] |
| **Musculoskeletal and connective tissue** | Muscle weakness | | | 5 | 2[b] |
| | Arthralgia | | | 4 | |
| **Nervous system** | Headache | 1 | | 2 | |
| | Aphasia | 1 | | 2 | |
| | Dysphasia | 2 | | | |
| | Paresthesia | | | 2 | |
| | Visual Disturbance | 1 | | | |
| **Respiratory** | Cough | | | 2 | |
| **Skin and subcutaneous tissue** | Rash | | | 3 | |
| | Pruritis | | | 4 | |

[a]One patient developed diarrhea requiring hospitalization after four cycles, which resolved with supportive care and was considered "possibly" pembrolizumab-related. A second patient developed diarrhea after five cycles, which resolved with dexamethasone and was considered "probably" pembrolizumab-related.

[b]One patient developed muscle weakness 5 weeks after starting pembrolizumab, requiring daily living assistance and high-dose dexamethasone–the SAE was considered "possibly" pembrolizumab-related. A second patient developed post-operative lateralized weakness, likely due to LITT.

$P = 0.001$) (Supplementary Fig. S1b); however, whether differences in tumor locations influenced these outcomes is unclear. To assess whether the survival advantage was confined to the early patients enrolled before the DSMC review, we compared Kaplan–Meier curves for the randomized and post-amendment expansion LITT + PEM cohorts; for both OS and PFS, these curves overlapped closely with one another and remained clearly separated from the NLS + PEM curve, indicating that the benefit was observed across early and later LITT + PEM patients (Fig. 4). Notably, *MGMT* promoter methylation did not affect survival in any cohort (Supplementary Fig. S2). In multivariable Cox hazard models adjusted for *MGMT* status, age, sex, *IDH* mutation, KPS, enrolling center, and number of prior recurrences, treatment arm was the only consistent independent predictor of both OS and PFS in the per-protocol population (Supplementary Table S2).

Best response analysis showed that in the LITT + PEM arm, nine patients (27%) had partial response (PR) and 14 (42%) had stable disease (SD) (Fig. 2b, c), while none achieved a complete response (CR), and ten patients (30%) had progressive disease (PD) including four with early progression (i.e., baseline MRI was the only obtained imaging). The resulting overall response rate (CR + PR) and disease control rate (CR + PR + SD) were 27% and 70%, respectively. The maximum duration of response was 32.8 months (from best response to progression) (Supplementary Table S3). In contrast, all six NLS + PEM patients experienced PD (Fig. 2).

In summary, patients treated with LITT plus pembrolizumab experienced significantly improved OS and PFS, as well as higher response and disease control rates, compared to those undergoing NLS plus pembrolizumab or those receiving LITT followed by a non-ICI therapy.

## LITT is associated with early activation of non-classical monocytes

LITT-generated hyperthermia induces immunogenic cell death, endothelial injury, and BBB disruption, especially in the non-ablative peritumoral region[20,21,24]. Because direct CNS sampling carries significant safety risks, we turned to peripheral immune profiling. Local reactivation of the GBM TME—such as via Tumor Treating Fields[30,31], LITT[21,27,28], or other localized immune-activating strategies[32]—has been shown to elicit robust responses detectable in PBMCs, which in turn serve as TME surrogates. Accordingly, serial PBMCs were collected from Phase 2 patients before NLS or LITT (Pre-Procedure), before cycle 1 (Post-Procedure), and before cycles 2, 4, 6, 9, 17, and 34 of pembrolizumab. For correlative analysis, we processed and analyzed only the University of Florida (UF) specimens to minimize batch effects and control costs and because five of the six evaluable NLS + PEM control patients were enrolled at UF. Seventeen UF patients (five NLS + PEM and 12 LITT + PEM) yielded 69 PBMC samples, generating 427,655 single cells by paired 5′ scRNA-seq and TCRα/β V(D)J sequencing. Single cells were clustered with Seurat[33,34], visualized via UMAP[35], resolving 32 immune cell types annotated as previously described[30,31], and T cells interrogated for TCR diversity using Simpson's index[36] (Fig. 5a).

Because LITT-induced tissue injury primarily triggers innate immune activation, we first examined NK cells, dendritic cells, and classical and non-classical monocytes in PBMCs collected before and after LITT (before C1 pembrolizumab). Although overall subset frequencies were unchanged, non-classical monocytes declined slightly (Supplementary Table S4). To assess functional shifts of non-classical monocytes in response to LITT, we applied the optimal transport algorithm Earth Mover's (Wasserstein) Distance Metric, a machine-learning approach adept at detecting subtle transitions in cell state[37,38], and observed a marked post-LITT transition of non-classical monocytes into subcluster 10—a change that was absent after NLS (Fig. 5b). Non-classical monocytes, which generally acquire migratory, phagocytic, and antigen-presenting functions in response to tissue injury, including that caused by LITT, can infiltrate damage sites to promote repair and adaptive immune activation[39,40]. Among the 12 LITT + PEM patients showing a shift into subcluster 10, we computed $\log_2$ fold-changes between pre- and post-LITT samples in non-classical

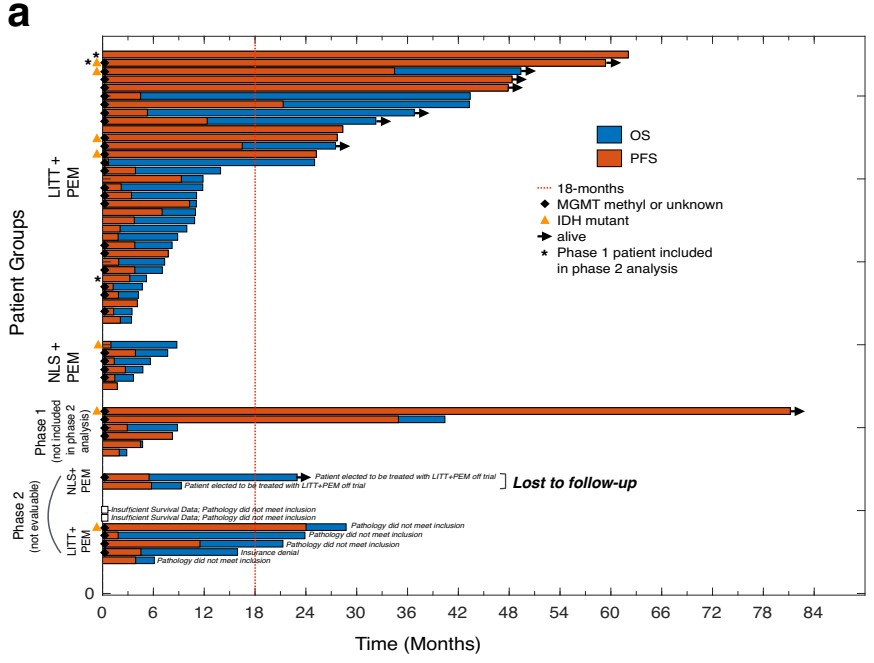

| Best response per RANO Per protocol n = 39 | NLS+PEM n (%) | LITT+PEM n (%) |
|---|---|---|
| Complete Response (CR) | 0 (0) | 0 (0) |
| Partial Response (PR) | 0 (0) | 9 (27.3) |
| Stable Disease (PD) | 0 (0) | 14 (42.4) |
| Progressive Disease (PD) or baseline MRI only | 6 (100) | 10 (30.3) |
| Overall Response Rate (CR + PR) | 0 (0) | 9 (27.3) |
| Disease Control Rate (CR + PR + SD) | 0 (0) | 23 (70) |

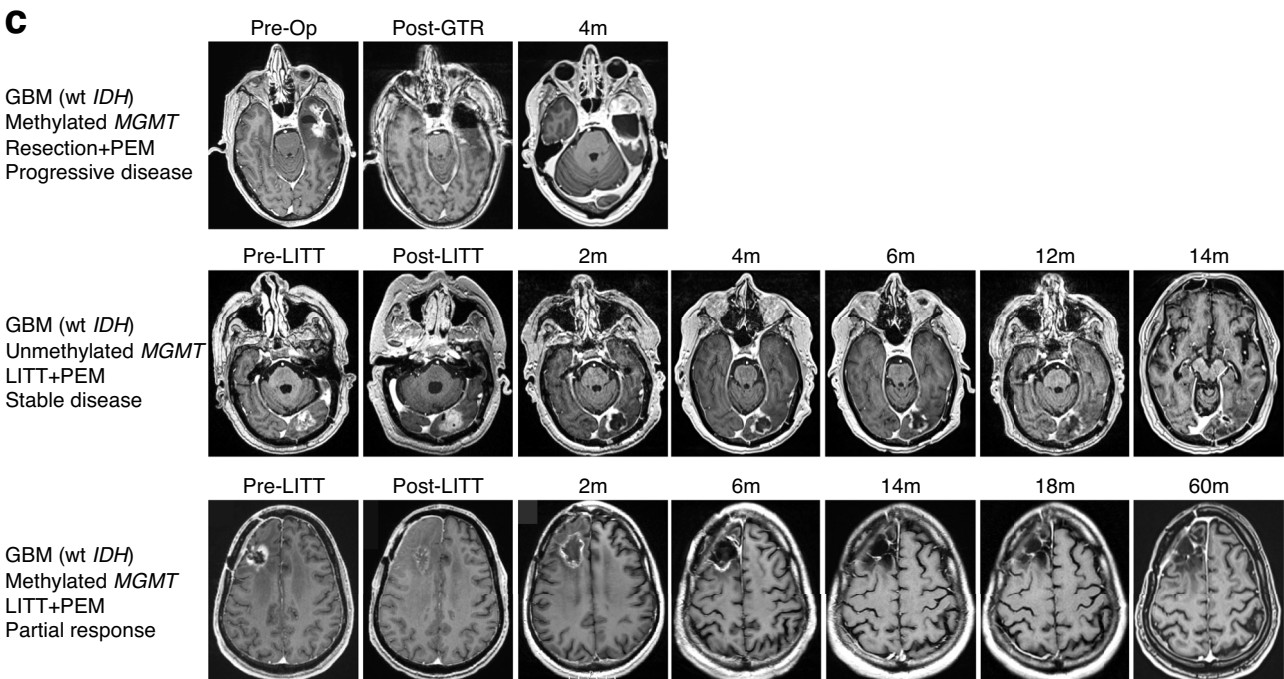

**Fig. 2 | Treatment, survival, and response summary. a** A swimmer plot for all enrolled patients. Phase 1 enrolled nine patients; three treated at the RP2D of pembrolizumab were included the per-protocol analysis. The per-protocol population comprised 39 patients: 33 received LITT + PEM (including the three phase 1 RP2D patients) and six received NLS + PEM. Seven patients were not evaluable per protocol (LITT + PEM, n = 5; NLS + PEM, n = 2). **b** Summary of best responses per RANO of 39 per-protocol patients. **c** Representative contrast-enhanced MRI examples showing early progressive disease (PD) in an NLS (Gross total resection) + PEM patient, prolonged stable disease (SD), and a partial response (PR) per RANO criteria in LITT + PEM patients. Pre-Op Pre-Operative, Post-GTR Post-Gross total resection.

monocytes, selected the 500 most differentially expressed genes ranked by false discovery rate (FDR), and grouped them into five functional pathways, i.e., adhesion/migration, macrophage differentiation, T cell activation, metabolic processes, and inflammatory response, to derive a composite non-classical monocyte activation score. This signature stratified patients into five with low activity and below-mOS [LITT + PEM (<mOS)] versus seven with high activity and above-mOS [LITT + PEM (> mOS)] (Fig. 5c). Expression of this program was concentrated in subcluster 10 (Fig. 5d) and enriched in longer

survivors by gene set enrichment analysis (GSEA) (Fig. 5e), and in an age-adjusted Cox proportional hazard (Cox PH) model, higher composite activity remained independently correlated with improved survival (Fig. 5f). To assess which components of this program might underlie the survival association, we used the same five pathway groupings to derive pathway-specific activity scores and fit exploratory age-adjusted Cox models. Higher cell-adhesion, metabolic-process, and inflammatory-response activity were each and together significantly associated with improved survival, whereas T-cell activation

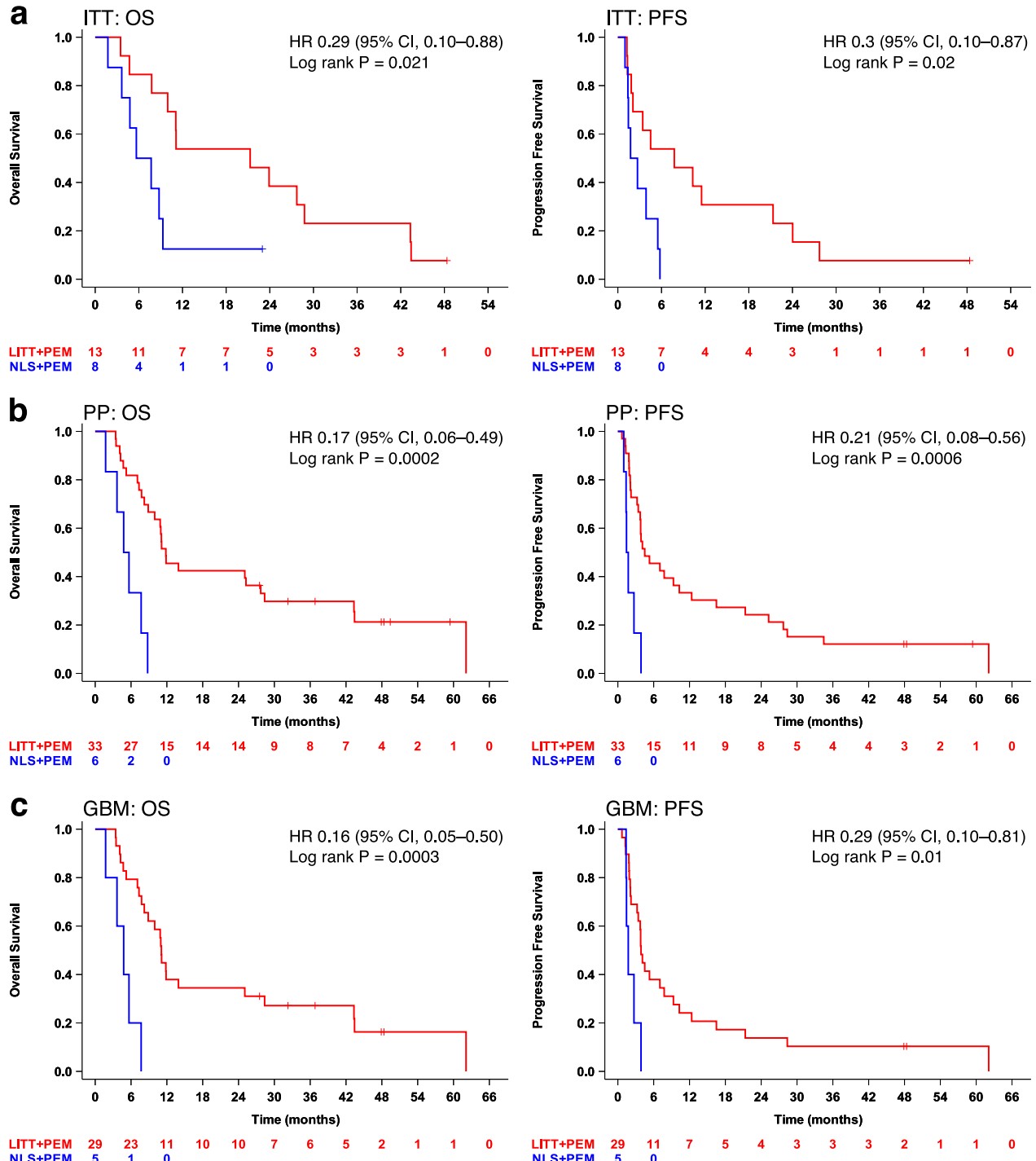

**Fig. 3 | Survival analysis. a** For the intent-to-treat (ITT) population (*n* = 21), mOS and mPFS were 21.3 (95% CI, 7.8–28.8) and 7.8 (95% CI, 1.8–21.3) months, respectively, in LITT + PEM versus 6.7 (95% CI, 1.7–9.3) and 2.2 (95% CI, 1.0–5.5) months in NLS + PEM. **b** For the per-protocol (PP) population (*n* = 39), mOS and mPFS were 11.8 (95% CI, 8.9–27.7) and 4.5 (95% CI, 3.4–10.3) months, respectively, in LITT + PEM versus 5.2 (95% CI, 1.7–N/A) and 1.6 (95% CI, 0.1–N/A) months in NLS + PEM. **c** For the wt-*IDH* GBM (GBM) population (*n* = 34), mOS and mPFS were 11.1 (95% CI

7.8–25.1) and 3.9 (95% CI 2.2–7.8) months, respectively, in LITT + PEM versus 4.8 (95% CI 1.8–N/A) and 1.7 (95% CI 1.4–N/A) months in NLS + PEM. Survival was measured from cycle 1, day 1 of pembrolizumab. Kaplan–Meier estimates were used for survival analysis, with comparisons made using the log-rank test, and hazard ratios (HR) with 95% confidence intervals (CI) were estimated using a Cox proportional hazards model.

showed a similar but non-significant trend, and macrophage differentiation was not correlated with outcome (Fig. 5f), suggesting that the survival signal captured by the 500-gene signature is largely accounted for by coordinated changes in adhesion, metabolic, and inflammatory programs within LITT-induced non-classical monocytes.

In short, local LITT specifically and acutely induces in the periphery a shift in non-classical monocytes to a new activated cell state, marked by high expression of pathways for migration, and metabolic and inflammatory processes–an effect not observed after NLS. This shift in cell state correlates with extended survival among LITT + PEM-

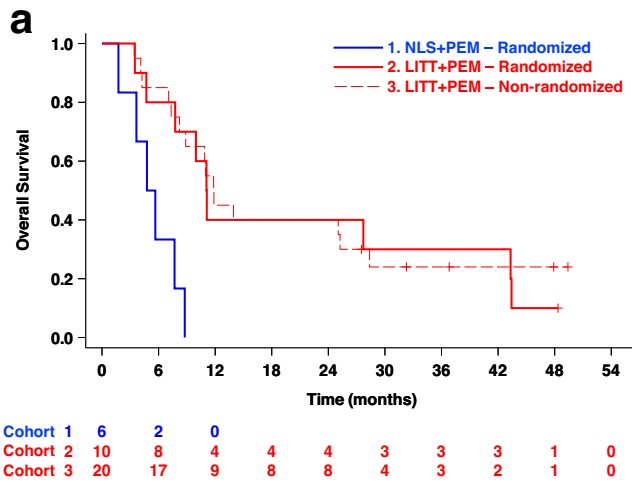

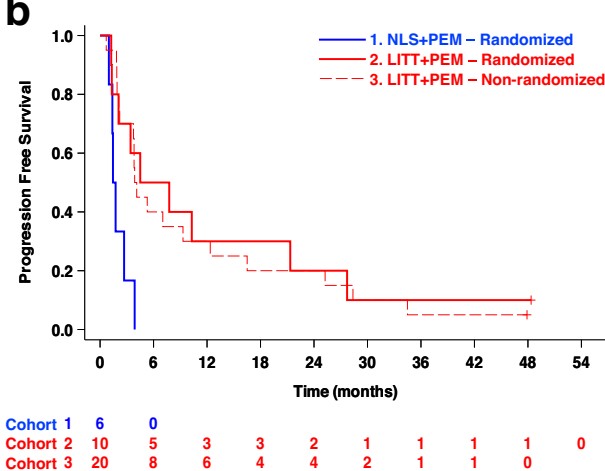

**Fig. 4 | Stability of LITT + PEM survival benefit across Phase 2 randomized and expansion per-protocol cohorts. a** Median OS was 5.2 months (95% CI, 1.7–N/A) in the randomized NLS + PEM cohort (Cohort 1; $n = 6$), 11.1 months (95% CI, 3.5–43.3) in the randomized LITT + PEM cohort (Cohort 2; $n = 10$), and 11.9 months (95% CI, 7.3–28.4) in the non-randomized LITT + PEM expansion cohort (Cohort 3; $n = 20$).

**b** Median PFS was 1.6 months (95% CI, 1.0–N/A) in the randomized NLS + PEM cohort, 6.1 months (95% CI, 1.2–21.3) in the randomized LITT + PEM cohort, and 4.0 months (95% CI, 2.1–12.4) in the non-randomized LITT + PEM cohort. Survival was measured from cycle 1, day 1 of pembrolizumab. Kaplan–Meier estimates were used for survival analysis.

treated patients and could serve as a candidate correlative biomarker of response.

### LITT + PEM is associated with heightened CD8⁺ T cell activation and clonal expansion

Next, we investigated whether LITT-induced innate activation could influence the immune response to pembrolizumab therapy. To this end, we extracted single T cell transcriptomes and corresponding TCRα/β V(D)J sequences from PBMCs gated on CD3, then sub-clustered them using Seurat, UMAP, and Simpson's diversity index[33–36], identifying 77,947 CD3⁺ cells belonging to 11 immunologically recognized T cell subtypes (Fig. 6a). In analyses of pre- vs. post-pembrolizumab (C1 vs C2) samples, cycling effector CD8⁺ T cells (*TYMS⁺GZMB⁺MKI67⁺*) in the LITT + PEM group showed markedly increased expression of both the 556-gene T cell activation pathway (GO: 0042110) and the 761-gene adaptive immune response pathway (GO: 0002250) (Fig. 6b, c), as well as a trend toward greater relative frequency (Supplementary Table S5), compared to the NLS + PEM group. Although most other T cell subtypes (both CD4⁺ and CD8⁺) did not differ significantly between these two groups (Supplementary Table S5), CD8⁺ T cells overall exhibited enhanced TCR clonal expansion, a hallmark of antigen-specific T cell activation[41], in LITT + PEM patients whose survival exceeded the median, distinguishing them from those with lower-than-median survival and from individuals in the NLS + PEM group (Fig. 6d). Notably, these findings align with the survival association previously observed for the LITT-activated non-classical monocytes, suggesting that early non-classical monocyte activation by LITT may foster an enhanced T-cell response upon subsequent pembrolizumab therapy.

### LITT + PEM induces a coordinated transition toward CD8⁺ effectors in long-term survivors

To investigate how CD8⁺ T cells progress in response to pembrolizumab following LITT, we examined their distribution across CD8⁺ T-cell subtypes in pre- and post-pembrolizumab samples from the same three patient groups: NLS + PEM, LITT + PEM (<mOS), and LITT + PEM (>mOS). At baseline (Pre-Procedure and C1), CD8⁺ T cells displayed largely overlapping distributions in all three groups. However, upon pembrolizumab administration, while all groups generated cycling T cells, the LITT + PEM (>mOS) group quickly diverged by C2, producing effector memory T cells that persisted throughout

treatment (Fig. 7a, red arrow). To follow the evolution of these CD8⁺ T-cell subtypes, we again applied the Wasserstein distance metric. In this analysis, arrows depict shifts between cell states, with arrow length and thickness representing the magnitude of movement and the number of cells in transit, respectively. In NLS + PEM patients, CD8⁺ T cells were relatively stagnant, showing minimal and uncoordinated evolution after NLS or pembrolizumab. In contrast, LITT + PEM patients exhibited more coordinated transitions into memory states, including central memory (CM), following LITT (Pre-LITT to C1), especially in those with survival exceeding the median. After initiating pembrolizumab (C1 to C2), both LITT + PEM subgroups began exiting memory states, but in longer-surviving patients, more cells progressed into the effector T cell cluster, whereas in shorter-surviving patients, cells shifted predominantly into exhausted states, including exhausted and stem-like exhausted clusters. With continued pembrolizumab exposure (C2 to C4), the shorter-surviving group showed further movement toward T cell exhaustion, whereas the longer-surviving group maintained a more robust effector phenotype throughout treatment (Fig. 7b).

To confirm that CM CD8⁺ T cells play a pivotal role in regulating response to LITT + PEM and survival, we tracked single TCR in all seven CD8⁺ T cell subtypes that originated from CM clones and followed their progression. In LITT + PEM (>mOS) patients, CM-derived cells transitioned into many other CD8⁺ subtypes following pembrolizumab (C1 to C2), resulting in fewer exhausted cells and more effectors. By contrast, in both NLS + PEM and LITT + PEM (<mOS) patients, CM-derived clones failed to regenerate a full range of subtypes but instead predominantly exhausted and stem-like cells (Fig. 8a). To verify that CM CD8⁺ T cells themselves drive these effects, we followed the transitions of CM cells directly using the Wasserstein distance metric. In LITT + PEM (>mOS) patients, CM clones displayed sustained proliferative capacity, evidenced by a significant, coordinated shift that generated progeny across the entire CD8⁺ T cell spectrum. In contrast, CM clones in LITT + PEM (<mOS) patients primarily transitioned into the exhausted states, and transitions in the NLS + PEM group were uncoordinated and largely random (Fig. 8b).

The results so far indicate that the CD8⁺ T-cell profile in longer-surviving LITT + PEM patients began to diverge from those in shorter-surviving patients primarily after the initiation of pembrolizumab. To investigate whether major genomic alterations might explain these

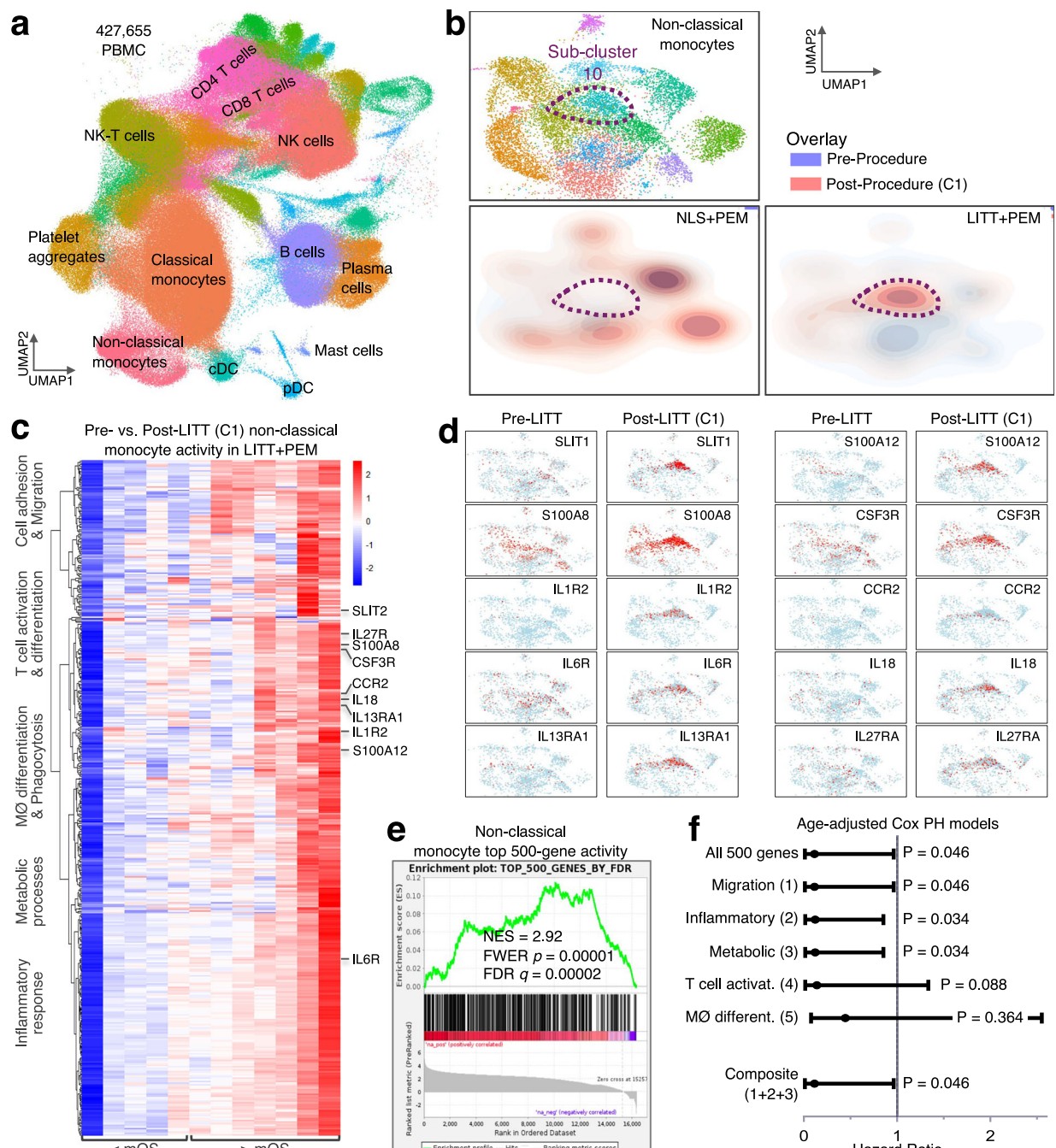

**Fig. 5 | Early activation of non-classical monocytes following LITT. a** UMAP (resolution 1) of 427,655 single PBMCs from 69 blood samples across 17 patients, showing 32 annotated immune subsets. cDC conventional DCs, pDC plasmacytoid DCs, NK natural killer cells, NKT NK T cells. **b** UMAP overlay of non-classical monocytes before (blue) and after (orange) LITT (before C1), with transition into subcluster 10 quantified by the Earth Mover's (Wasserstein) optimal transport metric (OT). The emergence of subcluster 10 post-LITT, absent in NLS, reflects a functional state shift detected by OT ($n = 17$). **c** Heatmap of $\log_2$ fold-changes for the top 500 FDR-ranked genes in non-classical monocytes (Pre vs. Post-LITT), ordered by mean fold-change from lowest to highest and clustered into five functional pathways as shown, which matches the LITT + PEM ( < mOS) ($n = 5$) and LITT + PEM (>mOS) ($n = 7$) cohorts. **d** UMAP highlighting post-LITT upregulation of genes linked to immune functions in the five pathways in (**c**) and concentrated in subcluster 10. **e** GSEA of Pre vs. Post-LITT fold-changes in the top 500 FDR-ranked genes in non-classical monocytes, demonstrating significant enrichment in subcluster 10 in LITT + PEM patients ($n = 12$). GSEA results are shown as signed NES (positive/negative enrichment). Significance (FDR q and family-wise error rate [FWER] p) is computed from one-sided, permutation-based tests in the direction of enrichment. **f** Age-adjusted Cox proportional hazards (Cox PH) models ($n = 12$) testing the association of overall survival with either the mean activity score of the top 500 FDR-ranked genes or the mean activity scores of the five functional pathways in (**c**), entered individually or jointly as indicated. Points show hazard ratios and lines 95% confidence interval (P-values are from two-sided Wald tests in R). Source data are provided with the bioinformatics codes.

differing outcomes, we obtained whole-exome sequencing from available baseline tumor samples, including four NLS + PEM, three LITT + PEM (<mOS), and four LITT + PEM (>mOS), to assess tumor mutational burden (TMB), a metric known to correlate with ICI responses[42]. No significant differences emerged in functional TMB, stop-gain or stop-loss single-nucleotide polymorphisms, or insertions/deletions across the three groups (Supplementary Fig. S3a). Similarly, there were no notable differences in the frequency of mutations,

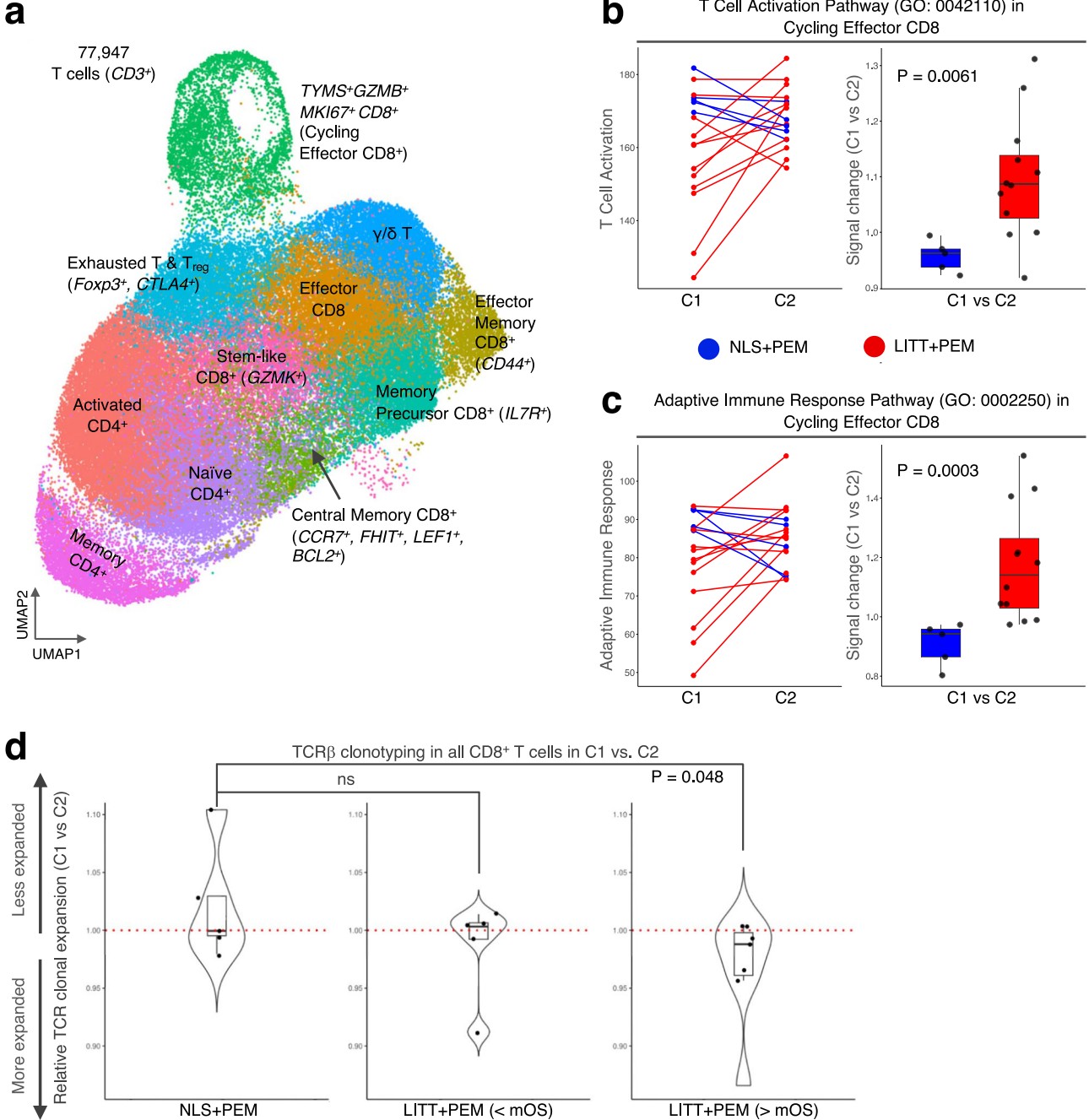

**Fig. 6 | LITT + PEM induces CD8⁺ T cell activation and clonal expansion. a** UMAP at resolution 1 of T cells annotated into 11 subtypes (*n* = 17). Left: Paired line plot of mean expression of the T-cell activation (**b**) and adaptive immune response (**c**) pathways at pembrolizumab cycles 1 (C1) and 2 (C2) in NLS + PEM (*n* = 5) and LITT + PEM (*n* = 12). Right: Combo box-and-whisker/dot plot of fold change in activation signals between C1 and C2 (C1/C2). **d** Combo violin/box-and-whisker/dot plots of TCRβ clonal diversity (Simpson's index) between C1 and C2 in NLS + PEM (*n* = 5), LITT + PEM (<mOS) (*n* = 5), and LITT + PEM ( > mOS) (*n* = 7), showing significant expansion in LITT + PEM (>mOS) patients. Data are represented as mean ± SEM. Whiskers indicate minimum and maximum values, boxes the interquartile range, and center lines the median. Statistical tests: two-sided paired Wilcoxon test (**b**, **c**) and two-sided paired Student's *t*-test (**d**). Source data are provided with the bioinformatics codes.

deletions, or amplifications in known canonical glioma-related genes (e.g., *CDKN2A/B, RB1, EGFR, PTEN, TP53*)[43] or in DNA excision repair and microsatellite stability genes (e.g., *MLH1, MSH2, POLE*)[44] (Supplementary Fig. S3b). Planned analyses of intratumoral PD-L1 expression could not be performed in a robust manner because only five tumors (three NSL + PEM and two LITT + PEM) had sufficient residual tissue; in this small subset, RNA-seq–derived PD-L1 levels did not differ appreciably between arms, and the sample size was too limited for any formal association with outcome (Supplementary Fig. S3c).

Next, to determine whether differential immune checkpoint expression might be correlated to observed outcome differences, we analyzed mRNA levels of *PD-1* and five additional checkpoint markers, including *CTLA-4, TIM-3, LAG-3, TIGIT*, and *IDO-1*, across all T cells at the first three timepoints (Pre-Procedure, Post-Procedure, or C1, and C2). All markers except *IDO-1* were comparably expressed and distributed among the three patient groups. By contrast, *IDO-1* initially showed elevated expression shortly after LITT within memory CD8⁺ T-cell populations (central memory, memory precursor effector, and

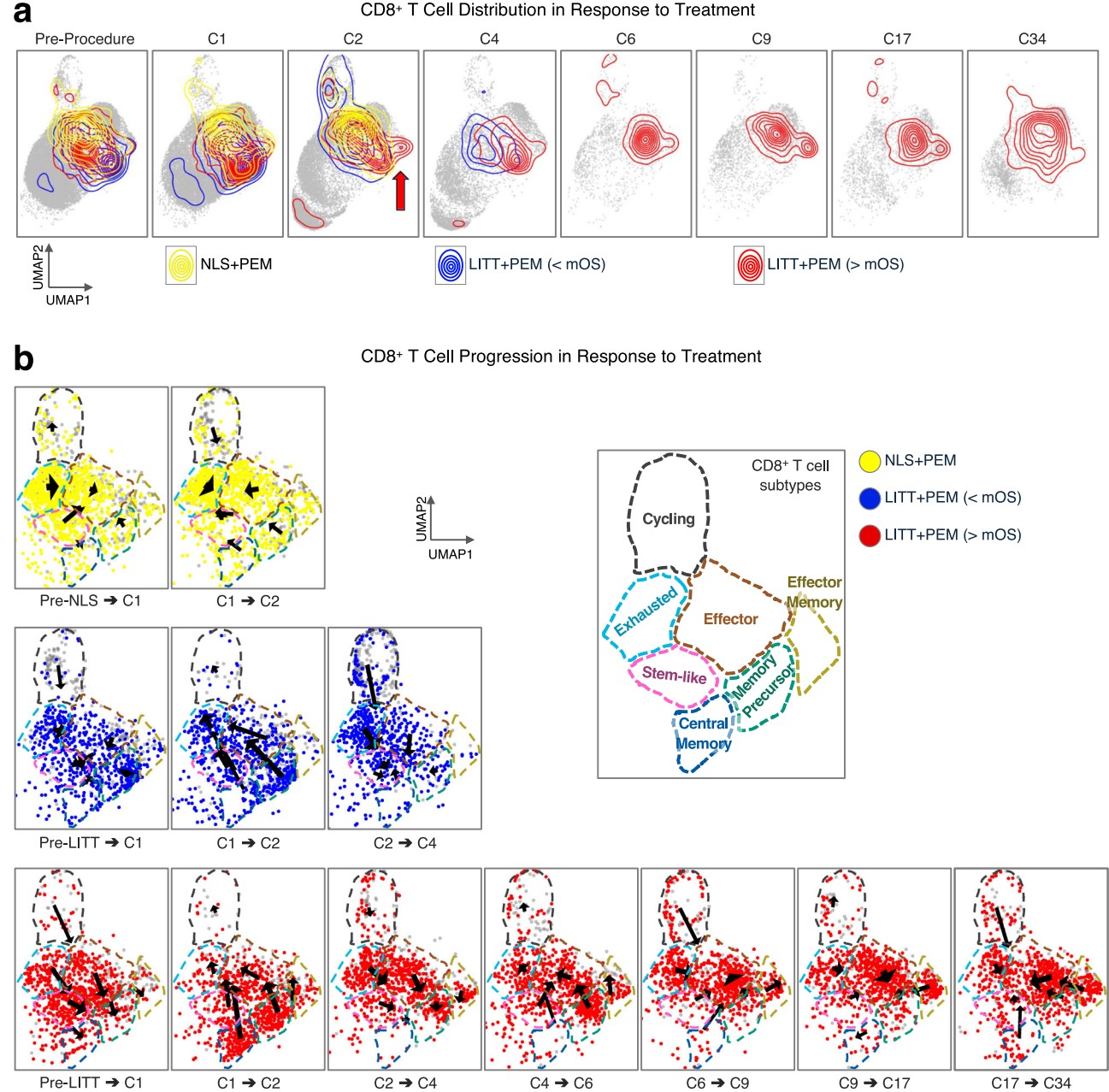

**Fig. 7 | Coordinated CD8⁺ T-cell transitions following LITT + PEM. a** 2D UMAP density plots of CD8⁺ T cells at pre-procedure, C1, and C2 timepoints for NLS + PEM, LITT + PEM (<mOS), and LITT + PEM (>mOS) groups. **b** UMAPs with arrows depicting transitions among seven CD8⁺ T-cell clusters, quantified by Earth Mover's (Wasserstein) OT distance. Arrow length indicates transition magnitude; thickness reflects cell counts. In long-surviving LITT + PEM (>mOS) patients, CD8⁺ cells shift coherently toward effector and memory clusters. In short-surviving LITT + PEM (<mOS), transitions favor exhausted phenotypes, while NLS + PEM controls show uncoordinated, random movement. $N$ = 5 for NLS + PEM; $n$ = 5 for LITT + PEM (<mOS); $n$ = 7 for LITT + PEM (>mOS). Source data are provided with the bioinformatics codes.

effector memory) in the LITT + PEM (>mOS) cohort (Fig. 8c), potentially reflecting the higher activation and proliferative capacity of these cells in that group. *IDO-1* encodes indoleamine 2,3-dioxygenase 1, an enzyme that regulates tryptophan metabolism; while transient increases in *IDO-1* can be an early indicator of T-cell activation, chronic upregulation can foster immunosuppression by limiting effector T-cell function and promoting regulatory T-cell activity[45,46]. Notably, upon pembrolizumab treatment, *IDO-1* expression was downregulated in most T cells, except for a small subset of cycling CD8⁺ T cells, in the LITT + PEM (>mOS) group, compared to its persistent expression across multiple T-cell clusters (particularly the exhausted subset) in the LITT + PEM (<mOS) and NLS + PEM groups (Fig. 8c, red arrow).

These findings suggest that the improved responses in longer survivors could be associated with LITT + PEM-driven downregulation of alternate immune checkpoints such as *IDO-1*.

Taken together, LITT plus pembrolizumab was safe and well tolerated and demonstrated better survival outcomes than NLS plus pembrolizumab in rHGA. Immune profiling revealed early non-classical monocyte activation, enhanced T-cell responses, and reduced *IDO-1* mRNA expression in those with prolonged survival. These findings suggest that localized thermal ablation may boost ICIs by creating a more favorable immune TME and mitigating key inhibitory pathways.

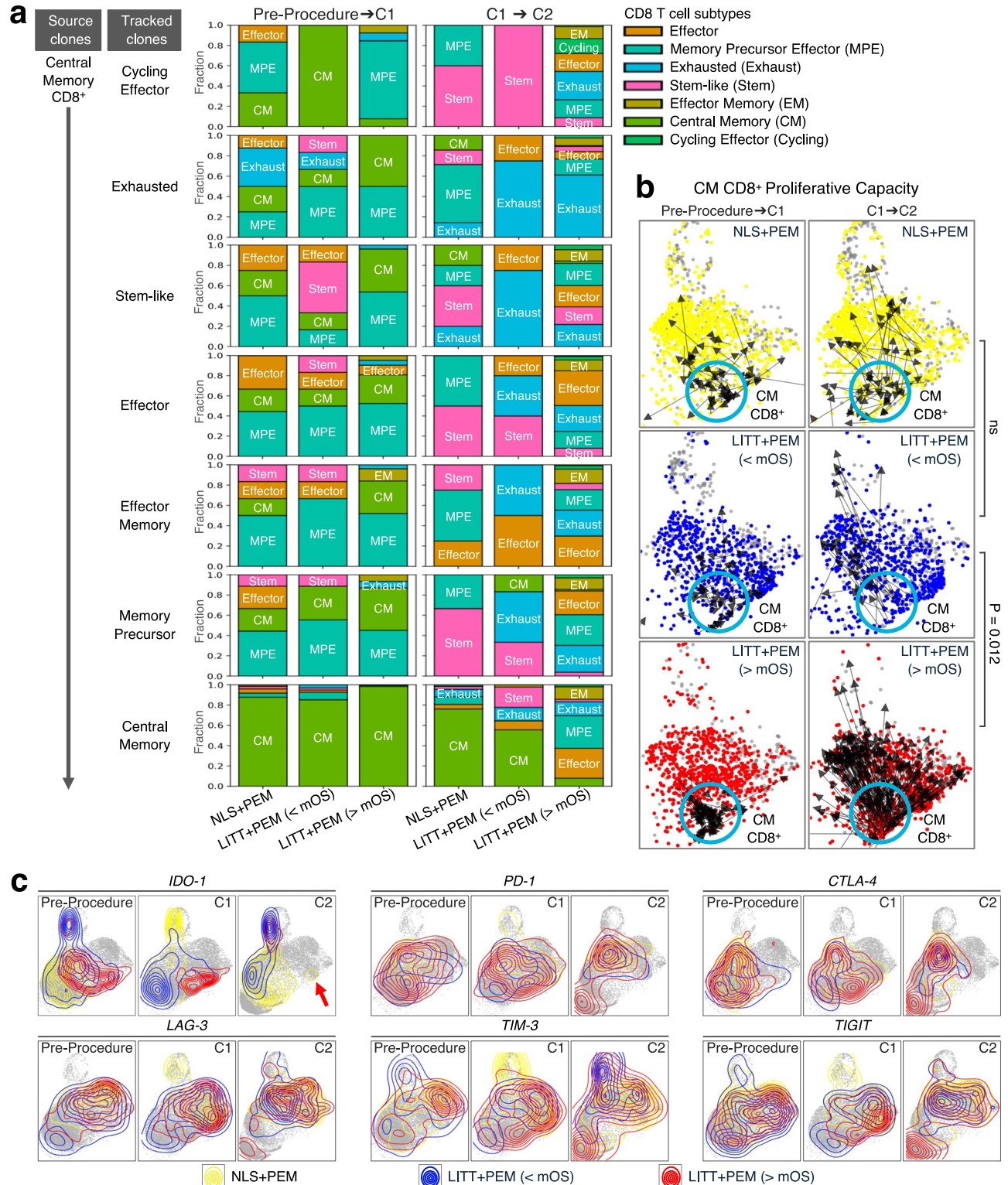

**Fig. 8 | Sustained CM CD8+ T-cell proliferation correlates with survival. a** CM CD8+ T-cell-derived source clones were tracked across Pre-Procedure→C1 and C1 → C2 transitions using the Wasserstein distance. Stacked bars show fractions of the seven CD8+ T-cell subtypes at endpoints in NLS + PEM, LITT + PEM (>mOS), and LITT + PEM (>mOS). Only in LITT + PEM (>mOS) do CM-derived clones repopulate all subtypes by C2. **b** Single CM CD8+ T cells (blue circle) were tracked across the same transitions. In LITT + PEM (>mOS), CM cells retain proliferative capacity, differentiating into all subtypes by C2. Clones are exhausted in LITT + PEM (<mOS) and uncoordinated in NLS + PEM. Chi-square statistics; *P* = 0.012; ns not significant. **c** UMAPs of immune checkpoints at Pre-Procedure, C1, and C2. LITT + PEM (>mOS) exhibit loss of IDO-1 expression in T cells (arrow). *N* = 5 for NLS + PEM; *n* = 5 for LITT + PEM (<mOS); *n* = 7 for LITT + PEM (>mOS). Source data are provided with the bioinformatics codes.

## Discussion

Our study provides evidence that combining localized LITT with systemic pembrolizumab can improve survival outcomes in patients with rHGA. This, alongside the immunological findings of early non-classical monocyte activation and robust T-cell responses, especially in long-term surviving LITT + PEM patients suggest that localized LITT effectively modifies the TME to potentiate the efficacy of ICIs. By enabling direct tumor cytoreduction, disrupting the BBB, and releasing neoantigens, LITT likely enhances the accessibility of cytotoxic T cells and pembrolizumab to the TME, thereby overcoming one of the primary barriers to effective immunotherapy in brain cancers. In-depth immune analyses from our study shed light on the systemic immunological changes associated with LITT + PEM. Early activation of non-classical monocytes following LITT may be integral for orchestrating a subsequent adaptive response, as evidenced by increased T-cell activation and clonal expansion after pembrolizumab initiation. Using a 500-gene non-classical monocyte activation signature, we found that higher activity was associated with improved survival. An exploratory pathway-level decomposition analysis suggested that this prognostic signal within LITT-induced non-classical monocytes is largely recapitulated by adhesion, metabolic, and inflammatory response programs, providing a focused mechanistic hypothesis that will require validation in larger cohorts, given the small sample size. A particular insight comes from observing how CM CD8+ T cells in LITT + PEM (>mOS) patients retain the capacity to generate the full spectrum of CD8+ T cell subtypes. In contrast, CM CD8+ T cells from shorter surviving patients and the NLS + PEM controls demonstrate a more limited transition, largely funneling into an exhausted T cell phenotype.

Pembrolizumab was selected as the PD-1 inhibitor primarily based on feasibility at the time of trial design, including an established systemic safety profile from studies in other solid tumors and an every-3-week dosing schedule that supported protocol adherence in a recurrent population. Experience with alternative checkpoint strategies in glioma was limited, and subsequent data have not demonstrated a survival advantage for intensified checkpoint blockade (e.g., adding CTLA-4 inhibition to PD-1 blockade) in recurrent GBM[4]. Emerging targets (e.g., LAG-3, TIM-3) remain investigational in this setting and warrant prospective evaluation, including in combination with LITT.

Although we found no major differences in TMB, canonical glioma-related mutations, or expression of several immune checkpoints in T cells when comparing the three patient groups, we observed that IDO-1 expression in T cells decreased primarily in the LITT + PEM group and was most evident in long-term survivors, whereas it remained higher across multiple T cell clusters in NLS + PEM and shorter-surviving LITT + PEM patients. This reduced expression in longer-term survivors may reflect less engagement of T cell exhaustion pathways beyond PD-1 inhibition, although a causal role for IDO-1 cannot be inferred from these data. This observation contrasts with the results of a Phase 3 trial in metastatic melanoma, where adding the IDO-1 inhibitor epacadostat to pembrolizumab did not extend pembrolizumab's benefit[47], highlighting the context-dependent complexity of IDO-1 biology across tumor types. Future investigations should incorporate longitudinal profiling of the tryptophan–kynurenine metabolic axis in T cells and plasma, as a functional readout of IDO-1 activity, together with single-cell and TCR analyses to test whether IDO-1–dependent metabolism influences transitions between exhausted and functional CD8+ T cell states or affects the differentiation potential of CM T cells under LITT-induced local and systemic conditions. These studies could illuminate how LITT plus ICI treatment reshapes the immune landscape in a way that may be more effective in GBM. Our findings also underscore the need for more comprehensive genomic and transcriptomic analyses, potentially through single-cell multi-omics or spatial transcriptomics, to uncover additional mechanisms of immune evasion. Such approaches would clarify how IDO-1 and other immunoregulatory genes interact with T-cell

differentiation states induced by LITT plus ICIs, while also accounting for epigenetic changes, sub-clonal driver mutations, and the spatial organization of immune cells. If IDO-1–dependent metabolism is confirmed to modulate durable benefit from LITT + PEM, combining PD-1 blockade with rational IDO-1–targeted agents may represent a more focused strategy than broadly escalating to multi-checkpoint combinations.

Despite these encouraging results, several limitations should be noted. First, randomization was halted early after an unscheduled DSMC review in the context of emerging data suggesting limited efficacy of ICI monotherapy in both this trial and the broader field at the time. This interim review was not prespecified and did not employ a formal alpha-spending plan; therefore, the overall type-I error rate for the randomized comparison is not controlled. Therefore, the observed survival differences should be interpreted as hypothesis-generating rather than definitive, even though survival curves in the subsequent non-randomized LITT + PEM expansion cohort are consistent with the initial signal. Consequently, the resulting sample size, particularly in the NLS + PEM arm, was relatively small, which may reduce the generalizability of the findings and diminish the statistical power to detect differences between groups. Nevertheless, the mOS and mPFS observed here in the NLS + PEM arm are consistent with outcomes reported in other ICI trials in recurrent GBM. In the CheckMate 143 study, which enrolled patients at first recurrence only, anti-PD-1 nivolumab monotherapy yielded mOS of nearly 10 months[4]. By contrast, all Phase 2 patients in our study were at least at a second recurrence, and approximately 10–15% were treated at a third recurrence, reflecting a more advanced and heavily pretreated population in which shorter survival would be expected. In addition, in the neoadjuvant pembrolizumab trial enrolling patients at first and second relapse, the post-surgical adjuvant pembrolizumab arm had a median OS of 6.4 months[11,12], similar to the approximately 5–6 month mOS observed in our NLS + PEM control arm. Taken together, these external data suggest that the observed survival signal with LITT + PEM is not readily explained solely by unusually poor outcomes in the NLS + PEM arm and is consistent with a genuine biological benefit. However, prospective randomized validation will be required to confirm efficacy. Further evidence comes from the extended survival seen in the LITT + PEM group compared with a CHC group receiving LITT followed by non-ICI therapy. While the reliance on historical data introduces potential biases and confounding factors, these data still provide some evidence for the conclusions drawn from our small, randomized cohort. In addition, variability in local treatment protocols, patient management, and correlative sample processing across sites may have introduced heterogeneity. To minimize these potential confounders, we focused our multi-omics correlative analyses on patients treated at a single institution, linking those profiles to survival outcomes within the same site.

Correlation of outcomes with intratumoral PD-L1 expression was limited by tissue availability, as LITT biopsies often yielded minimal residual material and PD-L1 testing was feasible in only a small subset of tumors; therefore, PD-L1-based analyses were not pursued. Notably, PD-L1 expression has been a modest and variable predictor of response to PD-1 blockade across tumor types[48,49]. Although PBMC-based profiling served as a practical surrogate for studying TME immune responses, direct interrogation of the intra-tumoral TME would offer deeper insights into immune cell functionality, spatial organization, and local cytokine dynamics. This is especially relevant for understanding how LITT disrupts both the tumor and peritumoral regions, potentially altering immune cell trafficking and activation in ways that peripheral sampling may underestimate. Nevertheless, repeated neurosurgical procedures pose substantial risks, making it challenging to obtain multiple intra-tumoral samples over time. A window-of-opportunity study design with carefully timed biopsies could help circumvent these limitations, facilitating a more comprehensive

evaluation of LITT-induced local immune responses while minimizing patient risk.

## Methods

### Study design and treatments

**Approval.** Human subject work was performed in accordance with an approved protocol by the respective internal review boards at the Washington University in St. Louis School of Medicine, the University of Florida College of Medicine, and the University of Southern California Keck School of Medicine, in accordance with the Declaration of Helsinki. A written informed consent was obtained from each human participant before study procedures and analysis were performed. All enrollments were completed at Washington University and the University of Florida. The correlative analysis was performed at the University of Southern California. The study's ClinicalTrials.gov identifier number is NCT02311582.

**Study design.** The study began with a Phase 1 lead-in to assess pembrolizumab safety in rHGA, followed by a randomized Phase 2 at a ratio 1:1 comparing NLS + PEM to LITT + PEM in rHGA. Randomization was computer-generated using a blocked scheme generated in SAS (v9.3+) and conducted centrally via the electronic data capture system RED-Cap. After emerging data demonstrated the limited benefit of ICI monotherapy in recurrent GBM (ClinicalTrials.gov, NCT02337491)[5], the independent Data and Safety Monitoring Committee (DSMC) requested an unscheduled interim review of accumulating efficacy data. On the basis of this review, the DSMC recommended closing the NLS + PEM arm and redirecting all subsequent enrollment to LITT + PEM (Protocol Amendment 12). Patients assigned to LITT + PEM could undergo optional surgical debulking prior to LITT; if resection was performed, the laser ablation then targeted only the residual tumor. To bolster our control framework and isolate LITT's effects without ICI, we then introduced a contemporaneous historical control (CHC) cohort treated with LITT plus non-ICI therapies (LITT+Non-ICI; Table 1). The primary interventions were LITT or NLS followed by adjuvant pembrolizumab, initiated within 1-week post-procedure (Fig. 1b). Patients receiving at least one dose of pembrolizumab were evaluable for survival and response analyses.

**Patient population.** Between May 2016 and May 2017, nine patients with radiographically recurrent WHO grade 3 or 4 astrocytoma were enrolled in the Phase 1 dose-escalation (3 × 3) cohort, receiving pembrolizumab at 100 mg, 150 mg, or 200 mg IV every 3 weeks following LITT (three patients per dose level). In Phase 2, from January 2017 to December 2022, 45 patients with histologically confirmed recurrent WHO grade 4 astrocytoma enrolled: 21 in the randomized cohort and 24 assigned to LITT + PEM after Amendment 12 (Fig. 1a). Key inclusion criteria were unequivocal tumor progression by RANO, a minimum of 12 weeks from frontline chemoradiation, a candidate for LITT or surgical resection, and Karnofsky performance status (KPS) ≥ 60%. Patients with suspected secondary WHO grade 4 astrocytoma—those with prior grade 2 or 3 tumors treated with chemoradiation, now with radiographic changes suggestive of transformation to grade 4—were conditionally enrolled and received pembrolizumab only after histopathological confirmation of grade 4 post-LITT or post-NLS. Key exclusion criteria included dexamethasone >4 mg/day at registration, prior immunotherapy (anti-PD-1/PD-L1 or anti-CD137), and bilateral multifocal or leptomeningeal disease. *IDH* mutation and multiple prior recurrences were permitted in both phases (Table 1).

**Contemporaneous historical controls.** We identified contemporaneous control patients treated at Washington University School of Medicine between 2016 and 2022 for histologically confirmed rHGA via retrospective chart review (see Supplementary Fig. S1a for the schema and criteria). Inclusion criteria mirrored those

of the prospective study and required biopsy-confirmed recurrence. We excluded patients with newly diagnosed grade 4 astrocytoma, those enrolled in other LITT trials, and individuals receiving immunotherapy. Of 122 rHGA patients treated with LITT during this period, 13 met all eligibility criteria.

**Laser interstitial thermal therapy.** LITT is an FDA-cleared, minimally invasive technique for thermal ablation of brain lesions. In our study, we employed the Monteris NeuroBlate Laser Ablation System (Minnetonka, MN) under MRI guidance. Following a small scalp incision and cranial burr hole, an MRI-compatible laser probe is advanced into the tumor. The system then delivers thermal energy, raising the lesion core temperature to ≈70 °C to induce coagulative necrosis, while peritumoral temperatures gradually climb to a peak of 40–45 °C, thereby limiting collateral injury[18–21].

**Statistical analysis plan for clinical and survival comparisons.** Phase I: A traditional 3 + 3 design was used to determine the recommended Phase 2 dose of pembrolizumab when combined with LITT for the treatment of rHGA.

Phase 2: The primary endpoint was progression-free survival (PFS); secondary endpoints were overall survival (OS), safety, objective response rate, and immune-response signatures. In both the randomized intent-to-treat and per-protocol cohorts, PFS and OS were measured from first pembrolizumab dose to radiographic progression (per RANO) or death. For the contemporaneous historical control (LITT+non-ICI), both endpoints were calculated from the date of LITT. Patients lost to follow-up or alive at data cutoff were censored at their last known alive date or at cutoff, whichever came first.

Initial Phase 2b calculations (one-sided $\alpha = 0.15$) assumed six-month PFS of 40% in the NLS + PEM arm—based on published rates (<10–30%) for common salvage regimens such as bevacizumab, temozolomide, carboplatin, carmustine, lomustine, etoposide, irinotecan[50–54]—and 65% in LITT + PEM, requiring 17 patients per arm for 80% power; we thus targeted 20 per arm. Randomization was not stratified by *IDH* mutation, *MGMT* promoter methylation, or other baseline factors because this planned small sample size did not permit reliable balance across multiple strata. After 21 randomized patients, emerging data showing limited ICI monotherapy benefit prompted the independent DSMC to request an unscheduled interim review of the accumulating trial data. This interim analysis was not prespecified in the original protocol and did not use a formal alpha-spending or stopping boundary. Descriptive comparisons at that time indicated longer PFS and OS in the LITT + PEM arm, and, together with the external evidence, led the DSMC to recommend closing the NLS + PEM arm and continuing the trial as a single-arm LITT + PEM extension in the spirit of minimizing harm, potential or perceived, to subjects and to complete the study in a timely manner. Under Amendment 12, the control arm closed, and the trial continued as a single LITT + PEM cohort. For this extension, we conservatively used a 2.9-month median PFS for the NLS + PEM arm (the upper 95% confidence bound rather than the 2.4-month median reported)[11]. Under that assumption, enrolling 27 additional patients (to achieve 20 evaluable GBM cases) provided 80.9% power (two-sided $\alpha = 0.05$, one-sample log-rank) to detect an increase in median PFS to 5.1 months versus the 2.9-month benchmark.

In the final analysis, we descriptively compared the PFS distribution of the per-protocol LITT + PEM group, estimated by Kaplan–Meier with its 95% confidence interval, to this 2.9-month benchmark. Two-sample log-rank tests comparing LITT + PEM with NLS + PEM in the randomized ITT and per-protocol populations, and with the contemporaneous LITT+non-ICI cohort, were performed as exploratory analyses to contextualize the magnitude of effect and do not preserve the nominal type I error. Kaplan–Meier curves for OS and PFS were generated with log-rank tests for these unadjusted comparisons.

Univariate and multivariable Cox proportional-hazards models adjusted for age, sex, *IDH* mutation, *MGMT* methylation, enrolling center, baseline KPS, and number of prior recurrences provided hazard ratios (HR) with 95% confidence intervals and *p*-values. All tests were two-sided at $\alpha = 0.05$. Analyses were performed in SAS v9.4 (Cary, NC).

### Tumor tissue processing and analysis

All available tumor samples from the study participants that remained after clinical diagnostics were prioritized for whole-exome sequencing and, when sufficient material was left, bulk RNA-seq. Exploratory analyses of intratumoral PD-L1 expression by immunohistochemistry were prespecified but could not be completed in a sufficient number of cases because most biopsies were very small and were exhausted by diagnostic testing and other genomic assays. Only five tumors (three NLS + PEM and two LITT + PEM) had adequate material for bulk RNA-seq–based estimation of PD-L1 expression.

### Single cell RNA-seq (scRNA-seq) analysis of PBMCs

Portions of the following scRNA-seq methods are adapted from our prior work[30], published under a Creative Commons license, CC BY 4.0, and are included here in full for completeness:

**Sample processing.** Cryopreserved PBMCs obtained from the study participants were rinsed in PBS, and cell viability was assessed using Trypan Blue staining. Single-cell suspensions were then prepared and applied to the Chromium Single Cell Chip (10x Genomics) as per the instructions provided by the manufacturer. Subsequently, single-cell RNA-seq libraries were generated using the Chromium Next GEM Single Cell 5′ v2 (Dual Index). To ensure consistency, all patient samples and the corresponding libraries were processed simultaneously in a single batch. Sequencing of the single-cell libraries was performed on an Illumina NovaSeq system, utilizing an 8-base i7 sample index read, a 28-base read 1 for capturing cell barcodes and unique molecular identifiers (UMIs), and a 150-base read 2 for the mRNA insert.

**Data processing.** The main operations were performed using the Seurat R package (3.2.2)[33,34], unless otherwise stated. When option parameters for the function deviated from the default values, we provided details of the changes accordingly. Most of the changes to the default options were made to accommodate and leverage the large size of the dataset. Cell Ranger Aggregation: The raw sequencing data were processed using cellranger mkfastq and cellranger multi commands of the Cell Ranger package as described in the TCR clonotyping section. Results from all libraries and batches were pooled together using the command cellranger aggr without normalization for dead cells, as it will be handled downstream. The filtered background feature barcode matrix obtained from this step was used as input for sequential analysis. Normalization of UMI: Using the global scaling normalization method, the feature expression for each cell was divided by the total expression, multiplied by the scale factor (10,000), and log transformed using the Seurat R function NormalizeData with method "Log Normalize". Seurat aggregation and correction for batch effect: As the counts were from two different batches, to align cells and eliminate batch effects for dimension reduction and clustering, we adopted the multi-dataset integration strategy as previously described[34]. Briefly, "anchors cells" were identified between pairs of datasets and used to normalize multiple datasets from different batches. We chose a reference-based, reciprocal PCA variant of the method detailed in the Seurat R package[33,34]. First, the previously integrated dataset was split by batches using the Seurat function SplitObject. Next, for each split object, variable features were selected using the function FindVariableFeatures. Features for integration were selected using the function SelectIntegrationFeatures, and PCA performed for each split object on the selected features. The anchor cells were identified by using the function FindIntegrationAnchors with the reference chosen as the largest among two batches and the reduction option set to "rpca". Finally, the whole datasets from the two batches were reintegrated using the function IntegrateData with the identified anchor cells.

**Uniform manifold approximation and projection (UMAP) dimension reduction.** The integrated multiple batch dataset was used as input for UMAP dimension reduction[35]. The feature expression was scaled using the Seurat function ScaleData, followed by a PCA run using the function RunPCA (Seurat) with the total number of principal components (PC) to compute and store option of 100. The UMAP coordinates for single cells were obtained using the RunUMAP function (Seurat) with the top 75 PCs as input features (dims = 1:75) with min.dist = 0.75 and the number of training epochs n.epochs = 2000. Clustering of cells: We relied on a graph-based clustering approach implemented in the Seurat package, which embeds cells in a K-nearest neighbor graph with edges drawn between similar cells and partitions nodes in the network into communities. Briefly, a Shared Nearest Neighbor graph was constructed using the FindNeigbhors function with an option dimension of reduction input dims = 1:75, error bound nn.eps = 0.5. This function calculates the neighborhood overlap (Jaccard index) between every cell and its k.param nearest neighbors[55]. The graph was partitioned into clusters using the FindClusters function with different values for the resolution parameter. The differential expressed gene markers for each cluster were found using the FindAllMarkers function with the option of only returning positive markers and a minimal fraction of cells with the marker of 0.25. The default Wilcoxon Rank Sum test was used to calculate statistical differences in each cell cluster. Alternative 2D embeddings, including t-SNE, were evaluated using quantitative metrics of within-cluster compactness, between-cluster separation, k-nearest-neighbor purity, silhouette width, and kNN-graph modularity, and UMAP most faithfully preserved the predefined clusters and trajectories; UMAP embeddings were therefore used for visualization and as the feature space for downstream distance-based analyses.

**Cell type annotation.** To delineate specific cell types within the data, cell type labels were assigned manually to clusters emerging from UMAP analysis. This annotation process was guided by the expression profiles of a set of marker genes, which are characteristic of various cell types including T cells, B cells, natural killer (NK) cells, monocytes, dendritic cells (DCs), myeloid-derived suppressor cells (MDSCs), megakaryocytes, red blood cells (RBCs), *CD34*+ stem cells, granulocytes, lymphocytes, macrophages, basophils, eosinophils, and neutrophils. The marker genes utilized for this purpose encompassed a wide range of immune response and cell differentiation indicators such as *CD3D, CD3E, ID3, IL7R, CCR7, ITGB1, CD95, TCF7, CD3D, CD3E, CD4, S100A4, CCR10, FOXP3, IL2RA, TNFRSF18, IKZF2, CTLA4, IL2, IL4, IL13, IL17A, CD3D, CD3E, CD8A, CD8B, CCL4, GZMA, GZMB, GZMH, GZMM, GZMK, IFNG, GNLY, TNF, PDCD1, LAG3, CD79A, CD79B, CD19, JCHAIN, GNLY, NKG7, CD16, NCAM1, KIR2DL4, SIGLEC7, CD14, LYZ, S100A8, S100A9, LGALS3, FCN1, FCGR3A, MS4A7, FCER1A, CST3, ITGAM, ITGAX, CLEC10A, CLEC9A, THBD, CD1C, LILRA4, CLEC4C, TLR7, TLR9, ITGAM, CD33, CD3D, CD3E, CD14, CD19, FUT4, CEACAM1, HLA-DRA, HLA-DRB1, HLA-DRB5, PPBP, PF4, ITGA2B, ITGB3, PEAR1, CD42D, CD59, HBG1, HBG2, HBB, CD34, CCR3, CD11b, CD13, CD18, CD229, CRACC, CD14, CD68, CD36, CD164, LAMP1, CD44, CD69, EMR1, MPO, CD62L, CD3D, CD3E, CD4, CD8A, CD8B, NKG7, GNLY, CD14, LYZ, FCER1A, CLEC10A, LILRA4, CLEC4C, CD79A, CD79B, HBB, PPBP,* and *PF4*. T cells were further divided into clusters to annotate subpopulations: naive CD4, central memory CD4, central memory CD8, anergic CD4, activated CD4, Treg, exhausted CD4, stem-like CD8, NKT, exhausted CD8, effector CD8, naive CD8, cytotoxic CD4, and effector memory CD8 using the following marker genes: *CD3D, CD4, CD8A, CTLA4, PDCD1, TIGIT, FOXP3, CCR7, GZMK, GZMB, GZMH, IL7R, CCL5, KLRB1, TRAV16, TRAV17, CX3CR1, CCL4, TRDC, CD69, FOS, BATF, IL2RB, TBX21, EOMES,*

*PRDM1*, and *ICOS*. Monocyte cells were further divided into classical and non-classical monocytes using the following marker genes: *CD14*, *IRF8*, *CCR2*, *CX3CR1*, and *FCGR3A*.

**Differential expression analysis.** To evaluate transcriptomic changes in non-classical monocytes between the Pre-LITT and C1 timepoints from the LITT + PEM cohort, a differential expression (DE) analysis was performed using an aggregated "pseudo-bulk" RNA-seq approach, followed by heatmap visualization of the top differentially expressed genes:

- **Pseudo-bulk aggregation:** the Seurat object was subset to retain non-classical monocyte cells at the Pre-LITT and C1 timepoints. For each patient and timepoint, the raw count data were summed across all cells in that group, yielding a pseudo-bulk expression profile (*genes × samples*). By using this aggregation strategy, cell-level biases are circumvented, and an approximation of bulk RNA-seq at the patient–timepoint level is achieved.
- **Differential expression with edgeR:**

  a. A DGEList object was created from the aggregated counts.
  b. Normalization factors were computed via calcNormFactors.
  c. A design matrix was constructed ($\sim Patient + TimePoint$) to compare Pre-LITT vs. C1.
  d. After dispersion was estimated, a negative binomial model was fit, and likelihood ratio tests (glmLRT) were performed.
  e. The resulting DE table was filtered for significance, capturing $\log_2$ fold change ($\log_2$FC) and FDR values.

- **Top-gene selection and visualization:** The top 500 differentially expressed genes by FDR were selected for further inspection. To highlight variation among patients, the $\log_2$FC (Pre-LITT vs. C1) were transformed into z-scores, and these scaled values were visualized in a heatmap. The heatmap columns were ordered according to patient survival times, thus facilitating a direct comparison of expression patterns across different clinical outcomes.

**Z-score calculation and GSEA ranking.** In the non-classical monocytes of the Experimental group, gene expression differences between time point Pre-LITT and time point C1 were assessed using $\log_2$FC values. For each gene, the mean $\log_2$FC was calculated separately for the short-term group ($\bar{x}_S$)—LITT + PEM (<mOS)—and for the long-term group ($\bar{x}_L$)—LITT + PEM (>mOS). The sample standard deviations (SD) were estimated for each group, and the number of patients in each group was denoted as $n_S$ (short-term or <mOS) and $n_L$ (long-term or > mOS).

A z-score was then computed to represent the difference between the two groups using the following equation:

$$z = \frac{\bar{x}_L - \bar{x}_S}{\sqrt{\frac{SD_L^2}{n_L} + \frac{SD_S^2}{n_S}}}, \qquad (1)$$

where $\bar{x}_L - \bar{x}_S$ indicates the difference in mean $\log_2$FC [(>mOS) minus (<mOS)], and the denominator represents the standard error. In this formulation, positive z-scores are indicative of higher expression in LITT + PEM (>mOS) survivors, whereas negative z-scores are indicative of higher expression in LITT + PEM (<mOS) survivors. In addition, genes with missing values or undefined z-scores were removed, and the remaining z-scores were compiled into a two-column rank file (gene name and z-score). The file was then sorted in descending order by the z-score and used as input for a GSEA.

**GSEA for pathway identification and confirmation.** Based on the z-scores computed as described above, a set of the top 500 differentially expressed genes by the lowest FDR was identified. This subset of genes, showing the most statistically significant differences between the Pre-LITT and C1 timepoints in non-classical monocytes in the LITT + PEM cohort, was treated as a custom gene set of interest. To further elucidate the biological pathways that may be driving survival differences, GSEA was performed using two inputs: (i) the ranked list of all genes (ordered by descending z-score), and (ii) the newly defined custom gene set of 500 genes. Following the standard procedure, GSEA was carried out by using the ranked list to calculate an enrichment score (ES), which reflects the degree to which the custom gene set is overrepresented at either the top (positive z-scores) or the bottom (negative z-scores) of the ranked list. Statistical significance was assessed by permuting the gene labels and recalculating the ES to generate a null distribution. This permitted estimation of the nominal *P*-value and the FDR *q*-value for enrichment.

**Cox proportional hazard model for survival prediction using the identified GSEA gene set.** The scaled and normalized $\log_2$FC of the top 500 genes by FDR was tested as a predictor of overall survival, beyond known clinical factors such as patient age, in a Cox proportional hazards model. First, the scaled and normalized $\log_2$FC of these 500 genes for each patient was summarized by taking the mean across the genes for each individual patient. Then, a binary covariate was created by categorizing patients into "high" or "low" expression groups based on the median of this summarized value for the cohort. This binary pathway-expression variable was combined with age (a continuous variable) as a covariate to determine whether patients with pathway-expression values above the median had a statistically different hazard ratio (i.e., risk of death or relapse) compared with those whose expression was below the median, after adjusting for age. The resulting hazard ratio and its confidence interval provided an estimate of how strongly pathway expression correlates with survival outcomes. The Schoenfeld residuals test[56] was used to assess the proportional hazards assumption in the Cox regression model, and no covariates demonstrated significant violation, indicating that the assumption held for the model.

**Pathway activity signal change comparison.** To compare pathway activity changes between NLS + PEM and LITT + PEM arms across consecutive timepoints, T-cell subpopulations were first defined based on clustering annotations derived from the Seurat workflow. For each pathway of interest (specified in a GMT file), the relevant genes were extracted, and the average expression of those genes computed within each patient and time point. Specifically, the following steps were performed:

- *Pathway Gene Extraction:* A GMT file containing predefined pathways was parsed to identify the set of genes corresponding to each pathway.
- *Subset and Normalization:* For each T cell subpopulation, a subset of the Seurat object was created. Expression values were normalized using the relative counts (RC) method (normalization.method = "RC" with a scale factor of $10^6$).
- *Mean Pathway Expression:* For each patient and time point, the mean expression of the pathway genes in the corresponding subpopulation was computed.
- *Ratio-Based Change Computation:* To quantify the relative fold change in pathway activity between two consecutive timepoints (A and B), the ratio of mean pathway expression at timepoint B to the mean pathway expression at timepoint A was used:

$$Signal\_Change = \frac{Mean\_Expr\_B}{Mean\_Expr\_A} \qquad (2)$$

- *Statistical Assessment:* For each comparison, the ratio values for the NLS + PEM and LITT + PEM arms were pooled, and a Wilcoxon rank-sum test was performed to evaluate differences between arms.

All comparisons were repeated across the T cell clusters of interest and for the two pathways, T Cell Activation (GO: 0042110) and Adaptive Immune Response (GO: 0002250), defined in the GMT file. Figures were generated to visualize both the per-patient longitudinal changes (line plots) and the distribution of ratio values by treatment arm (box plots).

## TCR clonotyping

Portions of the following TCR clonotyping methods are adapted from our prior work[30], published under a Creative Commons license, CC BY 4.0, and are included here in full for completeness:

**Sample processing.** Single-cell RNA-seq libraries were generated using the Chromium Next GEM Single Cell 5′ v2 (Dual Index) alongside the Human V(D)J Amplification Kit (10x Genomics), following the manufacturer's protocols. To ensure consistency, all patient samples and the corresponding libraries were processed simultaneously in a single batch. Sequencing of the single-cell libraries was performed on an Illumina NovaSeq system, utilizing an 8-base i7 sample index read, a 28-base read 1 for capturing cell barcodes and unique molecular identifiers (UMIs), and a 150-base read 2 for the mRNA insert.

**Data processing.** The 5′ single cell TCRα/β V(D)J library data were first processed using the 10x Genomics Cell Ranger package (v.7.0.0, with Java/9.0.1, bcl2fastq/2.20.0.422 dependencies). Command cellranger mkfastq was used to convert the raw sequencing data from the bcl to fastq format, and the cellranger multi command to align to the reference genomes GRCh38 (GENCODE v.24) and single cell clonal identification. Clonal tracking plots were created using the Immunoarch R package v0.9.0 (https://cloud.r-project.org/web/packages/immunarch/index.html) with the function trackClonotypes, option col = "a.a" to collapse all clones that share the same amino acid sequences.

**TCR clonotyping, diversity, and evolution.** A clonal tracking grid was established to map the presence and characteristics of TCRβ clones at each patient timepoint, focusing on aspects such as the number of cells, changes in cell type, and pathway activity.

- *Clonal Diversity by Simpson Index:* All unique TCRβ clones in the CD8 T cell subset were first identified for each patient. The Simpson diversity index, $D$, was then defined as:

$$D = 1 - \sum_{i=1}^{n} p_i^2, \tag{3}$$

where $p_i$ is the proportion of cells belonging to clone $i$, and $n$ is the total number of unique clones within a patient's T cell repertoire. Lower values of $D$ indicate reduced diversity (i.e., fewer clones dominate the repertoire), reflecting a higher degree of clonal expansion. Conversely, higher values of $D$ signify a more evenly distributed clonal landscape. Thus, this index measures the probability that two randomly selected individuals from the population will be of the same clone, with values ranging from zero (no diversity) to one (maximum diversity). This ratio provides a comparative measure of clonal diversity between two specified timepoints. A ratio greater than one indicates an increase in diversity, suggesting a diversification of the clonal population over time. Conversely, a ratio less than one reflects a decrease in diversity, pointing to a homogenization of the population. This analysis facilitates the understanding of clonal dynamics over the course of the study.

- *Statistical Comparison Between Cohorts:* After the Simpson index for each patient was computed, 2-sample $t$-tests were performed to compare the mean diversity between:

    NLS + PEM vs. LITT + PEM (<mOS).
    NLS + PEM vs. LITT + PEM (>mOS).

The corresponding $P$-values were calculated under the null hypothesis that the mean diversity of the two cohorts does not differ significantly. A low $P$-value ($P < 0.05$) indicates that one cohort exhibits a statistically different clonal diversity—and thus a different degree of clonal expansion—relative to the other.

**Tracking T cells Clones Across Timepoints.** An Optimal Transport (OT) algorithm[37,38] was used to trace CD8 T-cell clones between consecutive timepoints across three cohorts [NLS + PEM, LITT + PEM (<mOS), and LITT + PEM (>mOS)]. The analysis quantified how individual T-cell subpopulations (e.g., effector CD8⁺, exhausted CD8⁺ T cells) transitioned at both single-cell and cluster levels from one timepoint to the next. The steps were as follows:

- *Data Preparation and Subset Selection:* For each cohort and T cell subpopulation, single-cell expression and UMAP coordinates were collected at each available timepoint (e.g., Pre-Procedure, C1, C2, C4, etc.). Clusters of particular interest (COI) were merged where necessary to reduce sparsity.

- *Cost Matrix Construction for OT:* For each pair of consecutive timepoints, a cost matrix was computed based on Euclidean distances in principal component (PC) space. Let $x_i$ and $y_j$ be the PC embeddings for cell $i$ in the source timepoint and cell $j$ in the target timepoint, respectively, with the cost defined as:

$$C_{ij} = \| x_i - y_j \|_2. \tag{4}$$

Uniform distributions $a$ and $b$ were imposed over the source and target cells, respectively, ensuring equal weights for all cells.

- *Computing the OT Plan:* The OT plan $\pi^*$ was solved via the Earth Mover's Wasserstein Distance (ot.emd) function:

$$\min_{\pi \geq 0} \sum_{i,j} \pi_{ij} C_{ij} \, subject \, to \sum_{j} \pi_{ij} = a_i, \; \sum_{i} \pi_{ij} = b_j. \tag{5}$$

The resulting $\pi^*$ yields a principled "cell-to-cell matching" under global cost minimization.

- *UMAP-based Visualization:* Using the final OT plan, each source cell was mapped to a corresponding target cell, and arrows were plotted in UMAP coordinates:

Single-Cell Arrows: Each source cell had an arrow drawn to its corresponding target cell. Down-sampling was carried out to ensure consistent cell counts for plotting.
Cluster-Level Arrows: By aggregating source and target coordinates within each cluster, representative arrows were drawn for each cluster of interest.
Aggregated Arrow: Summed cluster displacements produced a single arrow reflecting the global population shift between two timepoints.

- *Stacked-Bar Distributions:* For each COI, the fractions of its cells in the source timepoint that mapped to each possible target cluster were computed. These fractions were represented as stacked bar charts, illustrating cluster transition patterns over successive timepoints.

- *Cohort Comparisons:* This pipeline was repeated for all T cell subpopulations across the three cohorts [NLS + PEM, LITT + PEM (<mOS), and LITT + PEM (>mOS)], enabling direct comparisons of clonal dynamics. Variations in both the directional shifts (arrows) and target cluster memberships (stacked bars) offered insight into how clonal populations evolved under different experimental conditions.

**Target distribution of T cell clones across timepoints.** Following the determination of how each source T cell mapped to a target cluster, cohort-specific target distributions were compared to assess whether different cohorts yield distinct post-mapping cluster compositions.

Specifically, for each cohort (e.g., NLS + PEM vs. LITT + PEM), the frequency with which source T cells, which were drawn from one timepoint and mapped to various clusters at the subsequent timepoint, was counted. These counts were then organized into a contingency table, where rows represented target clusters and columns represented cohorts.

Once the contingency table was constructed, a Chi-square test of independence was performed to determine whether the observed target cluster distributions differed significantly across cohorts. A low $P$-value (e.g., $P < 0.05$) indicated that the distribution of mapped target clusters was not independent of cohort membership, i.e., there was evidence to suggest that different cohorts lead to different cluster-level outcomes for T cells across timepoints. Conversely, a higher $P$-value implied that the observed distribution of target clusters could be attributed to chance alone, indicating no statistically significant difference between cohorts in their target distributions.

### Immune checkpoint expression comparison

To visualize immune checkpoint expression at multiple timepoints across three cohorts [NLS + PEM, LITT + PEM (<mOS), and LITT + PEM (>mOS)], a set of genes of interest (e.g., *PDCD1*, *CD274*, *IDO-1*) was defined, and FeaturePlot- and density-based visualizations were then generated as follows:

- *Data Preparation and Cohort Assignment:* Patients were first classified into three cohorts [NLS + PEM, LITT + PEM (<mOS), and LITT + PEM (>mOS)] based on OS. A Seurat object containing T-cell data was filtered to include only those patients overlapping with the survival dataset. The metadata was updated with a SurvivalGroup label for each cell, enabling straightforward subgroup analyses.
- *Timepoint Filtering and Down-sampling:* Analyses were restricted to timepoints Pre-Procedure, C1, and C2, discarding cells from other timepoints. To avoid uneven sampling, each cohort was down-sampled at each timepoint to ensure approximately equal cell counts, thereby reducing biases introduced by disproportional representation.
- *Split Panel FeaturePlots:* For each immune checkpoint gene, FeaturePlot was used to visualize expression on UMAP embeddings. The data were split by both cohort and timepoint (i.e., Group_TimePoint), resulting in subplots such as NLS + PEM_Pre-NLS, NLS + PEM_C1, and so forth. Each subplot included cells for the relevant subgroup and colored them according to the expression level of the target gene.
- *2D Density Plots for High Expressors:* In parallel, 2D density plots were generated to highlight cells with high (e.g., top 10% or 50%) expression of each gene. These top-expressing cells were colored by cohort, allowing comparisons of the spatial distribution of strongly expressing cells across cohorts within the UMAP space. Specifically, geom_density_2d was applied in ggplot2, with expression values used as weights, thus reflecting how each subgroup's high-expressing cells cluster or disperse.

### Reporting summary

Further information on research design is available in the Nature Portfolio Reporting Summary linked to this article.

## Data availability

The WES, bulk RNA-seq, scRNA-seq, and single-cell TCR sequencing data and associated clinical metadata generated in this study has been deposited in the European Genome-phenome Archive (EGA) under Study ID–EGAS50000001144 and Dataset ID–EGAD50000001639 [https://ega-archive.org/datasets/EGAD50000001639]. Access to these human genomics data is controlled to protect participant privacy and to ensure use consistent with informed consent and applicable ethics approvals. Access can be obtained by submitting a request

through EGA; requests are reviewed and access is granted upon approval by the study's lead contact (david.tran@med.usc.edu) and completion of the EGA Data Access Agreement. Additional individual de-identified participant data will be shared upon reasonable request. The study protocol is available in the Supplementary Information. The remaining data are available within the Article, Supplementary Information or Source Data. Source data are provided with this paper.

## Code availability

All bioinformatics codes with source data are described in the Methods section and can be accessed at https://github.com/TranLabUSC/MK_clinical_trial_paper (https://doi.org/10.5281/zenodo.18330074).

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

## Acknowledgements

This work was supported by a research funding and drug-provision grant from Merck and a research grant from Monteris. All aspects of the study, including conceptualization and design, data collection, analysis and interpretation, the decision to publish, and manuscript preparation, were conducted by the authors independently of the funders. We are grateful to our patients and their families for their participation. We also thank Jade Tao, Jingqin Luo, Grayson Talcott, Kaiden Barozinsky, Abeer Butt, and Ruth Katumba of Washington University in St Louis, Sonisha Warren, Annie Allen, Victoria Hope, Deborah Sampson, Valerie Greene Layton, Kristine Wynne, Renee Boyette, and Ciara Anderson of the University of Florida, and Sandra Oviedo of the University of Southern

California for study coordination and regulatory support. We appreciate Adam O'Dell and other members of the Tran laboratory of the University of Southern California for their insightful comments and feedback. Finally, we acknowledge the Siteman Cancer Center at Washington University in St. Louis, the UF Health Cancer Center, the Norris Comprehensive Cancer Center at USC Keck Medicine, and the USC Center for Advanced Research Computing for their support.

## Author contributions

J.L.C., A.P.G., and O.H.B. conducted the clinical study, interpreted the clinical results, and wrote the manuscript. H.M. and S.B.L. led the multi-omics correlative analysis, including designing and implementing all non-standard analytical pipelines specifically customized for this project, interpreted the correlative results, and wrote the manuscript. Y.X. and J.L. performed the statistical analysis of the clinical data and wrote the statistical section. D.C. assisted with the correlative analysis, annotated scRNA-seq clusters, analyzed the mutational data, interpreted the results. N.T. and W.A.L. provided technical assistance in sample processing, data collection, and analysis. M.R., T.J., G.A., G.P.D., J.H., J.L., A.Y.Z, and S.P. provided clinical expertise and feedback, participated in the clinical trial, and edited the manuscript. M.G.C., A.H.K, and E.C.L. contributed to the development and conduct of the study, interpreted the results, and wrote the manuscript. D.D.T. conceived the study, wrote the protocol, secured funding, conducted the clinical trial, directed and supervised the multi-omics correlative analysis, interpreted the results, and wrote the manuscript.

## Competing interests

D.D.T. has research funding from Merck, is a consultant for and received research funding from Monteris Medical, serves as a consultant for Novocure, and has research funding from Novocure, TVax, Novartis, Lacerta Therapeutics, Sarepta, Stemline, and Northwest Biotech for other unrelated work. A.P.G. has received personal honoraria for consultant work from Monteris Medical, Alexion Pharmaceuticals, Servier, ONO Pharma USA, Neosoma, Aptitude Health, Guidepoint, and Novocure. A.P.G. has held personal stock in Viatris Inc. O.H.B. serves on the advisory board for Novocure and as a consultant for Menarini. A.H.K. is a consultant for Monteris Medical and has research funding from Stryker to study a dural substitute, which has no direct relation to this study. J.L.C. received research fundings to support investigator-initiated studies from Merck, Incyte and GI Innovation, which have no direct relation to this study. E.C.L. owns stock in Neurolutions, Face to Face Biometrics, Caeli Vascular, Acera, Sora Neuroscience, Inner Cosmos, Aurenar, Petal Surgical, Cordance Medical, and Silent Surgical. He serves as a consultant for Monteris Medical, E15, and Neurolutions. He receives licensing fees from Neurolutions and Sora Neuroscience. Additionally, Washington University owns equity in Neurolutions. M.G.C. receives royalties from UpToDate. The remaining authors declare no competing interests.

## Additional information

Jian L. Campian[1,2,11,12], Son B. Le[3,4,12], Ashley Ghiaseddin[5,12], Omar H. Butt[1,2,12], Harshit Manektalia[3,4,12], Dongjiang Chen[3,4], Yifei Xu[6], Jingxia Liu[6], Maryam Rahman[5], Nathan Thai[3,4], William A. Leidig[2,7], Tanner Johanns[1,2], George Ansstas[1], Alice Y. Zhou[1], Sangami Pugazenthi[2,7], Gavin P. Dunn[8], Jiayi Huang[2,9], Milan G. Chheda[1,2] ✉, Albert H. Kim[2,7] ✉, Eric C. Leuthardt[2,7] ✉ & David D. Tran[3,4,10] ✉

[1]Division of Oncology, Department of Medicine, Washington University in St. Louis, St. Louis, MO, USA. [2]The Brain Tumor Center, Siteman Cancer Center, Washington University in St. Louis, St. Louis, MO, USA. [3]Division of Neuro-Oncology, Department of Neurological Surgery, Keck School of Medicine, University of Southern California, Los Angeles, CA, USA. [4]The Brain Tumor Center, University of Southern California, Los Angeles, CA, USA. [5]Department of Neurosurgery, University of Florida College of Medicine, Gainesville, FL, USA. [6]Division of Public Health Sciences, Department of Surgery, Washington University School of Medicine in St Louis, St. Louis, MO, USA. [7]Taylor Family Department of Neurosurgery, Washington University in St. Louis, St. Louis, MO, USA. [8]Department of Neurosurgery, Massachusetts General Hospital, Harvard Medical School, Boston, MA, USA. [9]Department of Radiation Oncology, Washington University in St. Louis, St. Louis, MO, USA. [10]Norris Comprehensive Cancer Center, University of Southern California, Los Angeles, CA, USA. [11]Present address: Department of Oncology, Mayo Clinic, Rochester, MN, USA. [12]These authors contributed equally: Jian L. Campian, Son B. Le, Ashley Ghiaseddin, Omar H. Butt, Harshit Manektalia. ✉e-mail: mchheda@wustl.edu; alberthkim@wustl.edu; leuthardte@wustl.edu; david.tran@med.usc.edu

