## [Transparent Peer Review file · Nature Communications]

Laser interstitial thermal therapy and adjuvant pembrolizumab in recurrent high-grade astrocytoma: a Phase 1/randomized Phase 2b trial

Corresponding Author: Professor David Tran

Version 0:

Reviewer comments:

Reviewer #1

(Remarks to the Author)

This study provides the first prospective evidence that LITT combined with PD-1 inhibition can extend survival and elicit systemic immune activation in recurrent high-grade astrocytoma. Despite the small control cohort and partially amended design, the consistency across clinical and immunologic endpoints strongly supports a biologic synergy between local thermal ablation and immune checkpoint blockade.

Interpretation of IDO-1 downregulation as mechanistic driver is intriguing but speculative.

-Frame this as an association rather than a causal link

-Discuss whether tryptophan–kynurenine metabolites were measured to support this.

Strong correlative claims (e.g., monocyte activation predicting survival) need caution given small n.

Explicitly state the DSMC's criteria for halting randomization (safety vs efficacy signal).

Consider adding sensitivity analyses (e.g., adjusted Cox models including center, KPS, recurrence number).

Clarify whether randomization was stratified by IDH or MGMT status.

(Remarks on code availability)

-

Reviewer #2

(Remarks to the Author)

The authors of the study present a Phase 1/2 multiinstitutional study of LITT and immunotherapy for recurrent high grade gliomas. The study demonstrates that LITT and Immunotherapy improves overall survival compared to the surgical resection and pembro group. The study demonstrates and OS of 11 months in the protocol group with a OS of 21 months in the ITT group. The results provided are encouraging and signal some change in the plasma biomarkers in T cells in response to LITT+ pembro group. There are a few significant limitations that should be addressed listed below that would enhance the paper, below:

1. The control surgery+ Pembro group outcomes are significantly worse than known contemporaneous control groups like Checkmate 143 (OS of 10 months). How is this explained? Would be good to compare their results to known trials in this setting to see if this is applicable.
2. The OS of ITT group is very promising but the OS is nearly 10 months different from the per protocol group. An explanation of this difference would be helpful. If there is a big outlier in the ITT group, it would be preferable to just put the protocol OS in the abstract so it is clear what the benefit of pembro and LITT is, without confounders
3. MRIs of the partial and stable response and treatment failures would be helpful. No imaging data or case data is presented in the main figures
4. A discussion on the use and picking of the Immune checkpoint inhibitor would be helpful as well. Is there a use for newer agents or PD-1, CTLA4, LAG3 inhibitors?

(Remarks on code availability)

No issues.

Reviewer #3

(Remarks to the Author)

While this manuscript described exciting new efficacious treatment of LITT + Pem in recurrent high-grade astrocytoma. However, I do have some concerns on the validity of the conclusion. The primary concern is that the study design and the statistical analysis are not aligned.

(1) Unplanned interim analysis: The original protocol did not include an interim analysis. Conducting an unplanned interim analysis can inflate the type I error rate because no pre-specified boundaries were defined. Without a pre-specified alpha-spending plan, it is unclear what p-value threshold was used to justify early stopping.

(2) Potential instability of early findings: Early trends in treatment effect are often unstable. The apparent superiority observed at an unplanned interim analysis may not have persisted had the trial continued, particularly when a liberal or post hoc boundary was used for decision-making.

(3) Mismatch between amended design and final analysis: After the interim analysis, the trial was amended to a single-arm design, referencing a historical median PFS of 2.9 months. The amended design and sample size calculation were based on a one-sample log-rank test. However, in the manuscript's analysis, two-sample log-rank tests were applied to both the per-protocol and randomized populations. This represents a clear inconsistency between the design and the analysis. If the analysis method differs from the design basis, it becomes uncertain whether the nominal power and type I error rate are still properly controlled.

(4) Was any analysis performed on intratumoral PD-L1 expression—either comparing expression levels between the two groups or evaluating the association between PD-L1 expression and PFS/OS? This analysis was described in the statistical analysis plan but was not reported or discussed in the manuscript.

(5) When you consider using UMAP to perform feature extraction, have you considered using some other common methods such as t-SNE? In addition, there might be some strong or weak associations, and as you have used the "FindNeighbors" function, I wish to know if you have found some structural clustering from the original sequencing. For example, some genes are highly correlated so that they potentially "work together". This is additional advice for further analysis. You may provide some insight if it takes too much time to re-run the analysis.

(6) For the Cox proportional hazard model that predicts the GSEA gene set, is there a mediation pathway for these genes to work? These gene sets may affect a specific biomarker, and further these biomarkers affect the survival probability. If so, would you be able to perform a mediation analysis?

(7) I am a bit confused with Figure 6 b, seems like only "pre-procedure to C1", and "C2 to C4" is shifting to memory precursor, and "C1 to C2" have some shift to exhausted and effector memory, would you be able to comment on these three.

(Remarks on code availability)

Reviewer #4

(Remarks to the Author)

(Remarks on code availability)

The code was not very clear/easy to follow since it integrates most of the formatting code and functional code. Lacks clear comments on how each function is performed and which figure/table it generates. There are a lot of lines of code that are commented out without a clear reason. Since this is not a methodological paper on a new algorithm or statistical innovation, I think it should suffice for a scientific research paper. I did not spend time re-running and checking all the code, but by roughly reading through, there seems to be no significant issue.

Version 1:

Reviewer comments:

Reviewer #1

(Remarks to the Author)

-

(Remarks on code availability)

-

Reviewer #2

(Remarks to the Author)

The authors have successfully addressed all of my comments. I believe this is worthy of publication. Congratulations.

(Remarks on code availability)

Reviewer #3

(Remarks to the Author)

The authors gave extensive responses and addressed our concerns.

(Remarks on code availability)

ReadME file is easy to understand and follow. I didn't install or run the code since it is too time consuming.

Reviewer #4

(Remarks to the Author)

(Remarks on code availability)

I have reviewed the updated code without running everything due to the anticipated time and effort needed. I have reviewed the structure of the guidelines for the analysis, which is well-written and easy to follow. I have reviewed most of the R scripts, and there are no obvious mistakes. The only suggestion from me is that the author may consider providing a small diagram in the GitHub page that illustrates the overall working process (trial design, data processing, sequencing, then UMAP, survival analysis, etc.), like what they did in the manuscript. This is not required for publication, just for the ease of future readers who wish to use the method.

We thank the reviewers for their careful evaluation and constructive comments, which have substantially improved our manuscript. In response, we have tempered causal interpretations (e.g., IDO-1 and monocyte activation), clarified key aspects of the trial design, interim analysis, and statistical methods (including new sensitivity and descriptive one-sample analyses), and explained the rationale for our choice of pembrolizumab and available biomarker data. We have also added representative MRI images, expanded and clarified our single-cell and computational analyses, and more fully acknowledged limitations and future directions.

Reviewers' comments are shown in black, with our responses in blue. Please note that all line-number references pertain to the Track Changes version of the revised manuscript.

Reviewer #1 (Remarks to the Author):

1. Interpretation of IDO-1 downregulation as mechanistic driver is intriguing but speculative.

-Frame this as an association rather than a causal link.

-Discuss whether tryptophan–kynurenine metabolites were measured to support this.

Our response: Thank you for this helpful comment. In the revised manuscript, we now describe IDO-1 downregulation as an association rather than implying a causal role. We did not measure tryptophan–kynurenine metabolites, as this was an unanticipated finding and we lacked sufficient remaining T cells or PBMCs from many subjects for metabolite analyses. We have added to the Discussion that future studies should directly evaluate this metabolic axis in T cells following LITT+PEM treatment to test this potential mechanism (Lines 561–566).

2. Strong correlative claims (e.g., monocyte activation predicting survival) need caution given small n.

Our response: Thank you for this comment. We have revised the text to present monocyte activation as a correlative finding rather than a strong predictor of survival (Lines 366–369).

3. Explicitly state the DSMC's criteria for halting randomization (safety vs efficacy signal). Consider adding sensitivity analyses (e.g., adjusted Cox models including center, KPS, recurrence number).

Our response: We have clarified in the main text and Methods that the DSMC's decision to halt randomization was driven by an efficacy signal, not by safety concerns. Specifically, in the context of emerging external data suggesting limited benefit of adjuvant immune checkpoint inhibitors alone, the DSMC requested an interim efficacy analysis per the monitoring plan. This analysis demonstrated a statistically significant survival advantage for the LITT+PEM arm compared with the NLS+PEM arm, leading the DSMC to recommend stopping further randomization (Lines 172–179; 958–963; 1031–1040). We have also added adjusted Cox sensitivity analyses (adding center, KPS, and recurrence number), which yielded results consistent with the primary analysis (Updated Supplemental Table S2) (Lines 294–297).

4. Clarify whether randomization was stratified by IDH or MGMT status.

Our response: Randomization was not stratified by IDH or MGMT status. The initial Phase 2b sample size calculations (one-sided $\alpha = 0.15$) assumed a 6-month PFS of 40% in the NLS+PEM arm, which was based on published rates (<10–30%) for common salvage regimens such as bevacizumab, temozolomide, carboplatin, carmustine, lomustine, etoposide, and irinotecan [1-5] and 65% in the LITT+PEM arm, requiring 17 patients per arm for 80% power. Given this relatively small, planned sample size, randomization was conducted without stratification factors, including IDH or MGMT status. We have added this clarification in the Methods sections (Lines 1031-1033).

Reviewer #2 (Remarks to the Author):

1. The control surgery+ Pembro group outcomes are significantly worse than known contemporaneous control groups like Checkmate 143 (OS of 10 months). How is this explained? Would be good to compare their results to known trials in this setting to see if this is applicable.

Our response: All patients enrolled in our Phase 2 trial were at least at their second recurrence, with 10–15% at their third recurrence, whereas CheckMate 143 included only patients at first recurrence [6]. Consequently, our cohort was more advanced and more heavily pretreated, which likely explains the shorter mOS observed in the NLS+PEM arm compared with the nivolumab arm in CheckMate 143. We consider the most appropriate comparison to be the post-surgical (adjuvant) PEM arm in the neoadjuvant pembrolizumab study [7, 8], which enrolled patients at first or second relapse. In that study, the adjuvant PEM arm had an mOS of 6.4 months, while our control arm showed an mOS of 5.3–6.2 months (PP vs ITT populations), indicating broadly comparable outcomes despite 15% of our patients being treated at a third, more advanced recurrence. We added this explanation to the Discussion section (Lines 614–634).

2. The OS of ITT group is very promising but the OS is nearly 10 months different from the per protocol group. An explanation of this difference would be helpful. If there is a big outlier in the ITT group, it would be preferable to just put the protocol OS in the abstract so it is clear what the benefit of pembro and LITT is, without confounders.

Our response: Thank you for this important observation. In the ITT population, 5 patients (2 in NLS+PEM and 3 in LITT+PEM) were included who were not evaluable in the per-protocol analysis because they did not complete protocol-defined treatment and/or have confirmed grade 4 astrocytoma. Given the small sample size, inclusion of these additional patients had a substantial impact and increased the mOS and mPFS estimates in both arms, which explains the ~10-month difference between the ITT and per-protocol results. There was no single extreme outlier driving the ITT findings. For transparency, we report both ITT and per-protocol outcomes and have clarified in the text (and abstract) that the per-protocol analysis represents the primary estimate of treatment effect, with the ITT analysis presented as a complementary sensitivity analysis. We also add this clarification to the Results section (Lines 54–59; 253–257).

3. MRIs of the partial and stable response and treatment failures would be helpful. No imaging data or case data is presented in the main figures

Our response: We have now added representative imaging data to **Figure 2, panel c**. Specifically, we include contrast-enhanced MRI examples illustrating (i) early progressive disease in a control-arm patient, (ii) prolonged stable disease, and (iii) a partial response as defined by RANO criteria.

4. A discussion on the use and picking of the Immune checkpoint inhibitor would be helpful as well. Is there a use for newer agents or PD-1, CTLA4, LAG3 inhibitors?

Our response: Thank you for this helpful suggestion. We have expanded the Introduction (Lines 118–122) and the Discussion (Lines 542–551) sections to clarify our choice of immune checkpoint inhibitor and to comment on newer agents. The study was designed in 2014, when CTLA-4 and PD-1/PD-L1 were the principal checkpoint pathways in clinical use or active clinical testing. We selected PD-1 blockade because several reports at that time showed that a substantial fraction of high-grade astrocytoma express PD-L1, that PD-L1 expression increases with WHO grade, and that PTEN loss (common in high-grade astrocytoma) further upregulates PD-L1 [9-13]. On this basis, and in consultation with Merck, pembrolizumab was chosen as the study drug; its every-3-week dosing schedule also reduced visit burden compared with every-2-week PD-1 regimens such as nivolumab and was considered favorable for patient adherence in this recurrent population. In contrast, ligands for CTLA-4 (CD80/CD86) are largely restricted to cells in the tumor microenvironment rather than tumor cells themselves, and subsequent data from CheckMate 143 demonstrated that adding ipilimumab to nivolumab did not improve efficacy but increased toxicity [14], so we did not pursue CTLA-4 combination therapy. At the time of trial development, newer checkpoints such as LAG-3 and TIM-3 were either in very early preclinical/clinical stages for brain tumors or not yet available. Looking ahead, if IDO-1 downregulation in T cells is validated as a key mechanism for durable responses to LITT+PEM, combining anti-PD-1 therapy with an IDO-1 inhibitor would be a rational strategy to explore in future trials (Lines 599–602).

Reviewer #3 (Remarks to the Author):

1. Unplanned interim analysis: The original protocol did not include an interim analysis. Conducting an unplanned interim analysis can inflate the type I error rate because no pre-specified boundaries were defined. Without a pre-specified alpha-spending plan, it is unclear what p-value threshold was used to justify early stopping.

Our response: We agree with this concern and now explicitly acknowledge that the interim analysis was not pre-specified and therefore does not control the overall type I error rate. The original protocol planned 1:1 randomization between MK-3475 alone (NLS+PEM) and MK-3475 after MLA (LITT+PEM). As external data emerged suggesting limited efficacy of post-surgical pembrolizumab alone (NLS+PEM), the DSMC requested an unscheduled interim review for ethical reasons. At that review, descriptive analyses showed higher PFS and OS in the LITT+PEM arm compared with NLS+PEM arm: in the LITT+PEM arm (2 Phase 1 and 10 Phase 2 patients), 6-, 9-, and 12-month PFS probabilities were 45.0%, 45.0%, and 33.8% with a median PFS of 5.49 months, and 6-, 9-, and 12-month OS probabilities were 75.0%, 62.5%, and 50.0% with median OS not estimable. In the NLS+PEM arm, the 3-month PFS probability was 44.4% with a median PFS of 2.73 months (95% CI: 1.41–3.88), and 3- and 6-month OS probabilities were 83.3% and 31.3% with a median OS of 4.77 months (95% CI: 1.74–8.78). Based on the magnitude and consistency of these differences, together with evolving external evidence, the DSMC recommended stopping randomization and converting the trial to a single-arm LITT+PEM study.

No formal alpha-spending plan or pre-specified p-value boundary was used; the decision relied on DSMC judgment in the spirit of minimizing harm, potential or perceived, to subjects and to complete the study in a timely manner. We have clarified this in the Methods (Lines 958–963; 1031–1040; 1056–1065) and Discussion (Lines 605–612), noting that the unplanned interim analysis is a limitation and that these results should be interpreted as hypothesis-generating. Full details of the interim analysis and statistical output are provided in the provided amended protocol (pp. 58–59).

2. Potential instability of early findings: Early trends in treatment effect are often unstable. The apparent superiority observed at an unplanned interim analysis may not have persisted had the trial continued, particularly when a liberal or post hoc boundary was used for decision-making.

Our response: We share this concern and, as outlined in our response to Comment 1, we now explicitly acknowledge that the interim analysis was unplanned, without a pre-specified alpha-spending boundary, and therefore the apparent early treatment effect should be regarded as hypothesis-generating and potentially unstable over time. Full details of the interim analysis and statistical output are provided in the amended protocol (pp. 58–59). To partially address the stability of the survival signal, we divided the per-protocol Phase 2 population into three cohorts: randomized NLS+PEM, randomized LITT+PEM, and non-randomized LITT+PEM (treated as an expansion cohort of the randomized LITT+PEM arm), and present their Kaplan–Meier curves together in the **new Figure 4**. The curves for the randomized and non-randomized LITT+PEM cohorts largely overlap and remain visually separated from the NLS+PEM control curve, suggesting that the survival advantage is consistent across early and later LITT+PEM patients. A similar approach of comparing randomized and subsequent expansion cohorts using overlaid K–M curves has been employed in a neoadjuvant pembrolizumab study in GBM published in *Nature Communications* [8], which we now cite as precedent. Nevertheless, we emphasize in the Discussion that, given the unplanned interim analysis and modest sample size, these findings require confirmation in larger, prospectively randomized trials.

3. Mismatch between amended design and final analysis: After the interim analysis, the trial was amended to a single-arm design, referencing a historical median PFS of 2.9 months. The amended design and sample size calculation were based on a one-sample log-rank test. However, in the manuscript’s analysis, two-sample log-rank tests were applied to both the per-protocol and randomized populations. This represents a clear inconsistency between the design and the analysis. If the analysis method differs from the design basis, it becomes uncertain whether the nominal power and type I error rate are still properly controlled.

Our response: We agree with this point and have revised the manuscript to better explain how the amended design relates to the final analyses. After the DSMC-mandated conversion to a single-arm study, the amended protocol specified a one-sample log-rank comparison of PFS in the LITT+PEM cohort against a historical median PFS of 2.9 months and used this framework for the sample size calculation. In the revised Statistical Methods and Results, we now explicitly describe this historical 2.9-month PFS as the design benchmark and state that, in the final analysis, the PFS distribution of the per-protocol LITT+PEM cohort was compared descriptively to this benchmark using Kaplan–Meier estimates and their 95% confidence interval, without performing a formal one-sample log-rank hypothesis test (Lines 172–179; 1031–1045).

The two-sample log-rank tests comparing LITT+PEM with NLS+PEM are now explicitly described as exploratory, secondary analyses intended to contextualize the magnitude of effect within the

randomized portion of the trial; we note that these post hoc comparisons were not part of the amended design and do not preserve the nominal type I error. As described in our response to Comment 2, we also include an overlaid Kaplan–Meier plot of three cohorts (randomized NLS+PEM, randomized LITT+PEM, and non-randomized LITT+PEM) in the **new Figure 4** to illustrate the consistency of the survival signal across early and later LITT+PEM patients.

4. Was any analysis performed on intratumoral PD-L1 expression—either comparing expression levels between the two groups or evaluating the association between PD-L1 expression and PFS/OS? This analysis was described in the statistical analysis plan but was not reported or discussed in the manuscript.

Our response: Thank you for highlighting this, and we apologize for not describing our PD-L1 analyses in the initial submission. As specified in the statistical analysis plan, we intended to compare intratumoral PD-L1 expression between treatment arms and to explore its association with PFS/OS. In practice, this was not feasible because tumor tissue was very limited, particularly in the LITT+PEM arm, where patients underwent biopsy only. Many biopsies were small, with scant viable tumor and suboptimal quality, raising a high risk of under-sampling.

Given these constraints, we prioritized assays according to tissue requirements and sensitivity (DNA-based, then RNA-based, then protein-based). All usable samples after clinical diagnostic needs were processed for whole-exome sequencing; of these, only 5 tumors (3 NLS+PEM and 2 LITT+PEM) had sufficient material left for bulk RNA-seq, and none had adequate remaining tissue for PD-L1 IHC as originally planned. In these 5 RNA-seq samples, we did not observe statistically significant differences in PD-L1 expression between arms (**new Supplemental Figure S3c**), but the sample size is far too small for any definitive conclusion. We have now added a description of these limitations and limited results to the Results (lines 479–483), Discussion/Limitations (Lines 643–647), and Methods (Lines 1067–1073) sections, and clarify that the planned PD-L1–survival correlation analyses could not be performed. Of note, recent reports have suggested that PD-L1 expression in the TME has only a modest correlation with response to anti-PD-1 inhibitor [15, 16].

5. When you consider using UMAP to perform feature extraction, have you considered using some other common methods such as t-SNE? In addition, there might be some strong or weak associations, and as you have used the “FindNeighbors” function, I wish to know if you have found some structural clustering from the original sequencing. For example, some genes are highly correlated so that they potentially “work together”. This is additional advice for further analysis. You may provide some insight if it takes too much time to re-run the analysis.

Our response: Thank you for this thoughtful suggestion. We do routinely examine alternative non-linear dimensionality reduction methods, including t-SNE. For this study, we selected UMAP because the low-dimensional embedding is used not only for visualization but also as a feature space for downstream quantitative analyses (e.g., computing Earth Mover’s Distance to track subtype changes over time), where preservation of both local and broader-scale structure is important.

To substantiate this choice, we performed a systematic comparison of UMAP and t-SNE embeddings for all PBMCs, T cells, and non-classical monocytes using the same clustering and several quantitative metrics (within-cluster compactness, between/within distance ratio, silhouette score, kNN purity, and 2D kNN-graph modularity). Across all three datasets, UMAP produced markedly more compact clusters, equal or better relative separation between clusters, higher silhouette scores and kNN purity, and higher modularity, indicating better alignment between the 2D geometry and biologically defined clusters. These results are summarized in a **new**

supplemental document for review only, entitled “Comparison of UMAP and t-SNE embeddings.” We also added a statement in the Method section the rationale of using UMAP (lines 1134–1138).

Regarding the “FindNeighbors” function: In our current analysis, the FindNeighbors function was used to construct a cell-cell graph and identify clusters of transcriptionally similar cells, which we then used for cell type annotation as presented in the manuscript. This procedure does not directly assess gene-gene correlations or infer gene interaction networks; rather, it operates at the level of cells. We agree that investigating structural relationships among genes (e.g., co-expression or gene-gene interaction patterns) could provide additional biological insight and could a valuable direction for future extensions of this study.

6. For the Cox proportional hazard model that predicts the GSEA gene set, is there a mediation pathway for these genes to work? These gene sets may affect a specific biomarker, and further these biomarkers affect the survival probability. If so, would you be able to perform a mediation analysis?

Our response: In our Cox proportional hazards model, the predictor is a composite activity score derived from the top 500 genes associated with non-classical monocyte activation (Figure 5c–f), rather than a single GSEA pathway or individual biomarker. Conceptually, our biological model is that LITT induces an early shift in non-classical monocytes, and this activation program in turn relates to survival; in that sense, the monocyte program can be viewed as a putative mediator of the LITT–survival relationship. However, given the small cohort size and the strong collinearity between different summaries of this program, we did not feel that a formal causal mediation analysis with explicit estimation of direct and indirect effects of treatment on survival would be stable or interpretable.

To address the spirit of your question, i.e., through what biology the 500-gene signal may act, we performed an *exploratory pathway-level decomposition* of this signature. Starting from the same 500 genes shown in Figure 5c, we annotated them into five functional pathways (cell adhesion, T-cell differentiation, macrophage differentiation, metabolic processes, and inflammatory responses) based on known gene functions and derived a pathway activity score for each patient. In age-adjusted Cox models that included one pathway score at a time, three of the five pathways (cell adhesion, metabolic processes, and inflammatory responses) each reproduced the survival association observed with the full 500-gene score, whereas T-cell differentiation showed a similar but non-significant trend and macrophage differentiation was not associated with outcome (updated Figure 5f).

We then asked whether these pathway-level signals contributed information beyond the broader 500-gene program. To do this, we performed a formal mediation analysis using Cox-based causal mediation and Fisher-exact A–M tests. Binary pathway scores were strongly associated with the 500-gene exposure and behaved as prognostic markers for OS when considered alone with age. However, once the global 500-gene signature was included, the pathways no longer provided robust independent prognostic information, and the estimated proportion of the effect mediated by any single pathway was small and non-significant. In the special case of the cell-adhesion pathway, the mediator and exposure are perfectly correlated, so the pathway could potentially stand in for the full 500-gene signature, but we cannot statistically separate their effects in this small cohort. Together, these findings suggest that the pathways are biologically meaningful sub-modules of the 500-gene program rather than distinct mediating mechanisms. The detailed

description of the formal mediation analysis is attached as a **supplemental file for review only, entitled “Mediation Analysis.”**

7. I am a bit confused with Figure 6 b, seems like only “pre-procedure to C1”, and “C2 to C4” is shifting to memory precursor, and “C1 to C2” have some shift to exhausted and effector memory, would you be able to comment on these three.

Our response: The pattern you note in Fig. 6b (now new Figure 7b) — with pre-procedure→C1 and C2→C4 transitions predominantly shifting toward memory-precursor states, whereas C1→C2 shows more movement into exhausted and effector-memory states — is consistent with a dynamic, wave-like response of CD8⁺ T cells to antigen release and PD-1 blockade. After LITT, tumor antigens are released into the circulation in intermittent “pulses” rather than as a single bolus, as we observed previously when measuring brain-resident antigens following LITT [17]. Each pulse of antigen exposure in theory can recruit and activate both pre-existing memory T cells and newly generated clones.

With each subsequent cycle of pembrolizumab, these antigen-experienced T cells expand further. Previously exhausted cells in particular can be reinvigorated and transiently acquire effector or effector-memory phenotypes, which explains the increased flux from central/stem-like memory into effector and effector-memory clusters in the C1→C2 interval. As antigen levels decline, a proportion of these cells then contract back toward memory and memory-precursor states, which is what we predominantly observe in the pre-procedure→C1 and C2→C4 transitions. Thus, the three intervals you highlight likely capture different phases of repeated activation–expansion–contraction cycles.

Using our Earth mover’s distance metric–based approach, we can quantify the net direction of these shifts among T-cell clusters. On average, in the LITT+PEM long-term survivors, CD8⁺ T cells cycled more between memory and effector states and transitioned less frequently into the exhausted state than in the shorter-surviving cohort across corresponding time points. Specifically, during the C1→C2 transition, arrows from central memory, effector-memory precursors, and stem-like cells tended to terminate within effector/effector-memory clusters in long-term survivors, whereas in short-term survivors, arrows from central memory and stem-like clusters preferentially converged on the exhausted cluster. Similar repeated waves of T-cell expansion and return to memory, coupled to ongoing TME activation and anti-PD-1 therapy, have also been reported with other TME-activating modalities, such as Tumor Treating Fields plus pembrolizumab [18] and with immune checkpoint blockade in other tumor types [19, 20], although those prior studies did not directly analyze cell-state transitions as we do here.

Reviewer #4 (Remarks to the Author):

Our response: Thank you for your time and helpful comments.

References for the responses to reviewer's comments:

1. Franceschi, E., et al., *Phase II trial of carboplatin and etoposide for patients with recurrent high-grade glioma*. Br J Cancer, 2004. **91**(6): p. 1038-44.
2. Reardon, D.A., et al., *Phase I trial of irinotecan plus temozolomide in adults with recurrent malignant glioma*. Cancer, 2005. **104**(7): p. 1478-1486.
3. Brandes, A.A., et al., *How effective is BCNU in recurrent glioblastoma in the modern era? A phase II trial*. Neurology, 2004. **63**(7): p. 1281-4.
4. Kappelle, A.C., et al., *PCV chemotherapy for recurrent glioblastoma multiforme*. Neurology, 2001. **56**(1): p. 118-20.
5. Chua, S.L., et al., *Phase 2 study of temozolomide and Caelyx in patients with recurrent glioblastoma multiforme*. Neuro Oncol, 2004. **6**(1): p. 38-43.
6. Reardon, D.A., et al., *Effect of Nivolumab vs Bevacizumab in Patients With Recurrent Glioblastoma*. JAMA Oncology, 2020. **6**(7): p. 1003.
7. Cloughesy, T.F., et al., *Neoadjuvant anti-PD-1 immunotherapy promotes a survival benefit with intratumoral and systemic immune responses in recurrent glioblastoma*. Nature Medicine, 2019. **25**(3): p. 477-486.
8. McFaline-Figueroa, J.R., et al., *Neoadjuvant anti-PD1 immunotherapy for surgically accessible recurrent glioblastoma: clinical and molecular outcomes of a stage 2 single-arm expansion cohort*. Nature Communications, 2024. **15**(1): p. 10757.
9. Wintterle, S., et al., *Expression of the B7-related molecule B7-H1 by glioma cells: a potential mechanism of immune paralysis*. Cancer Res, 2003. **63**(21): p. 7462-7.
10. Wilmotte, R., et al., *B7-homolog 1 expression by human glioma: a new mechanism of immune evasion*. Neuroreport, 2005. **16**(10): p. 1081-5.
11. Parsa, A.T., et al., *Loss of tumor suppressor PTEN function increases B7-H1 expression and immunoresistance in glioma*. Nat Med, 2007. **13**(1): p. 84-8.
12. Yao, Y., et al., *B7-H1 is correlated with malignancy-grade gliomas but is not expressed exclusively on tumor stem-like cells*. Neuro-Oncology, 2009. **11**(6): p. 757-766.
13. Bloch, O., et al., *Gliomas Promote Immunosuppression through Induction of B7-H1 Expression in Tumor-Associated Macrophages*. Clinical Cancer Research, 2013. **19**(12): p. 3165-3175.
14. Omuro, A., et al., *Nivolumab with or without ipilimumab in patients with recurrent glioblastoma: results from exploratory phase I cohorts of CheckMate 143*. Neuro-Oncology, 2017. **20**(5): p. 674-686.

15. Davis, A.A. and V.G. Patel, *The role of PD-L1 expression as a predictive biomarker: an analysis of all US Food and Drug Administration (FDA) approvals of immune checkpoint inhibitors*. *J Immunother Cancer*, 2019. **7**(1): p. 278.
16. Lu, S., et al., *Comparison of Biomarker Modalities for Predicting Response to PD-1/PD-L1 Checkpoint Blockade: A Systematic Review and Meta-analysis*. *JAMA Oncology*, 2019. **5**(8): p. 1195-1204.
17. Leuthardt, E.C., et al., *Hyperthermic Laser Ablation of Recurrent Glioblastoma Leads to Temporary Disruption of the Peritumoral Blood Brain Barrier*. *PLoS One*, 2016. **11**(2): p. e0148613.
18. Chen, D., et al., *Efficacy and safety of adjuvant TTFIELDS plus pembrolizumab and temozolomide in newly diagnosed glioblastoma: A phase 2 study*. *Med*, 2025.
19. Wang, K., et al., *Combination anti-PD-1 and anti-CTLA-4 therapy generates waves of clonal responses that include progenitor-exhausted CD8(+) T cells*. *Cancer Cell*, 2024. **42**(9): p. 1582-1597.e10.
20. Huang, A.C., et al., *T-cell invigoration to tumour burden ratio associated with anti-PD-1 response*. *Nature*, 2017. **545**(7652): p. 60-65.

Point-by-point response to the reviewers' comments (NCOMMS-25-76760B)

We thank the reviewers for their careful evaluation and enthusiasm for the manuscript. Reviewers' comments are shown in black, with our responses in blue.

Reviewer #1 (Remarks to the Author):

No comments provided

Our response: Thank you for your time.

Reviewer #2 (Remarks to the Author):

The authors have successfully addressed all of my comments. I believe this is worthy of publication. Congratulations.

Our response: Thank you for your time and helpful comments.

Reviewer #3 (Remarks to the Author):

The authors gave extensive responses and addressed our concerns.

Reviewer #3 (Remarks on code availability):

ReadME file is easy to understand and follow. I didn't install or run the code since it is too time consuming.

Our response: Thank you for your time and helpful comments.

Reviewer #4 (Remarks to the Author):

Reviewer #4 (Remarks on code availability):

I have reviewed the updated code without running everything due to the anticipated time and effort needed. I have reviewed the structure of the guidelines for the analysis, which is well-written and easy to follow. I have reviewed most of the R scripts, and there are no obvious mistakes. The only suggestion from me is that the author may consider providing a small diagram in the GitHub page that illustrates the overall working process (trial design, data processing, sequencing, then UMAP, survival analysis, etc.), like what they did in the manuscript. This is not required for publication, just for the ease of future readers who wish to use the method.

Our response: Thank you for your time and helpful suggestion. We have added a diagram in the GitHub page as suggested.